# Global implications of a low soil moisture threshold for microbial hydrogen uptake

Linta Reji [1,2,5] ✉, Matteo B. Bertagni [1,3,6], Fabien Paulot [4], Qianhui Qin [1,2] & Xinning Zhang [1,2] ✉

The impact of increasing anthropogenic hydrogen ($H_2$) emissions on Earth's radiative balance depends on the soil microbial $H_2$ sink—the largest and most uncertain term in the global $H_2$ budget. Soil moisture is a primary but poorly quantified control regulating the soil sink. Here, we assess the sensitivity of microbial $H_2$ oxidation to soil moisture in laboratory experiments with temperate and arid soils spanning distinct textures. We report $H_2$ oxidizer activity down to −70 to −100 MPa water potentials across soils, which are among the driest conditions reported for microbial activity and are much drier than assumed in global simulations of $H_2$. Using genome-resolved meta-omics, we link $H_2$ oxidation dynamics in temperate soils to specific desiccation-adapted microbial taxa that contribute differentially to $H_2$ uptake along the moisture gradient. Through global simulations, we show that our observationally constrained drier moisture threshold increases the contribution of arid and semi-arid regions for soil $H_2$ uptake by 4-7 percentage points (pp), while decreasing the contribution of temperate and continental regions (−7 pp). Our results highlight the importance of $H_2$ uptake under extreme hydrological conditions, particularly the roles of desertification, dryland expansion, and $H_2$-oxidizer ecophysiology in modulating long-term changes in $H_2$ uptake.

Molecular hydrogen ($H_2$) is an abundant trace gas in the troposphere, with a current global mean concentration of ~550 ppb in the marine boundary layer[1]. $H_2$ is expected to play a key role in the global clean energy transition due to its potential as a low-carbon fuel[2,3]. However, an increasing reliance on $H_2$ energy will likely raise atmospheric $H_2$ concentrations, as $H_2$ is leak-prone[4,5]. This warrants careful evaluation of the potential climate effects of increasing $H_2$ levels in the atmosphere. While $H_2$ is not a greenhouse gas, it still imparts a significant indirect radiative forcing due to its interaction with hydroxyl (OH) radicals in the atmosphere. This interaction prolongs the lifetime of tropospheric methane and contributes to the production of stratospheric water vapor and tropospheric ozone[6–11]. Given the significance of $H_2$ for Earth's radiative balance (100-year time-horizon Global

Warming Potential, GWP100, estimated to be 11.6 ± 2.8[11]), the balance of the global $H_2$ budget is a critical consideration in the context of accelerated energy transitions.

Uptake by soil microbes is the largest sink for atmospheric $H_2$, accounting for 60–80% of the global tropospheric $H_2$ sink[8,12–14]. By reducing the amount of $H_2$ available to react with OH radicals in the atmosphere, the soil sink considerably dampens the indirect radiative impact of $H_2$. Recent culture-based and genomic work has established that soil uptake is microbially mediated, with the high-affinity hydrogen oxidizing bacteria (HA-HOB) being widely distributed and active across ecosystems[15–21]. The magnitude of the global microbial $H_2$ sink is far from constrained and is the largest uncertainty in assessing the indirect global warming potential of $H_2$[11].

[1]High Meadows Environmental Institute, Princeton University, Princeton, NJ, USA. [2]Department of Geosciences, Princeton University, Princeton, NJ, USA. [3]Department of Civil and Environmental Engineering, Princeton University, Princeton, NJ, USA. [4]Geophysical Fluid Dynamics Laboratory, National Ocean and Atmosphere Administration, Princeton, NJ, USA. [5]Present address: Department of the Geophysical Sciences, The University of Chicago, Chicago, IL, USA. [6]Present address: Department of Environment, Land and Infrastructure Engineering, Politecnico di Torino, Torino, Italy. ✉e-mail: lreji@uchicago.edu; xinningz@princeton.edu

One of the primary sources of uncertainty in soil sink estimates is the lack of quantitative constraints on the effects of soil moisture variability on $H_2$ uptake[22–27]. This is due to the complex ways in which shifts in soil moisture affect $H_2$ uptake, including direct impacts on microbial metabolism and the regulation of $H_2$ diffusion into soil pores[24,26]. At the dry end of the moisture gradient, HA-HOB are water-stressed, resulting in limited $H_2$ oxidation due to biotic constraints. Uptake is also low at higher moisture levels, due to the reduced diffusion of $H_2$ in water-filled pores, a physical constraint on uptake. Thus, the relationship between soil moisture and $H_2$ uptake is highly non-linear[23,25,26].

Several attempts have been made to parameterize the soil sink to capture its sensitivity to soil moisture variability, along with other abiotic factors such as temperature and soil type[8,23,25,26,28,29]. Due to the paucity of observational constraints on soil uptake[25,30–34], particularly in moisture-limited, arid, and semi-arid soils, these parameterizations result in inaccurate estimates of the seasonality and spatial variability of uptake[8,27,35]. Most parameterizations include a water-stress threshold, below which uptake is essentially zero; however, the magnitude of this threshold is poorly constrained[8,29]. Similarly, an optimal moisture level at which peak uptake occurs has been identified as a key threshold regulating microbial $H_2$ oxidation, as above this level, uptake declines due to limited diffusion of $H_2$ and $O_2$[26]. The uncertainty associated with these moisture thresholds makes it challenging to represent the spatio-temporal variability in the $H_2$ sink strength in global models[27].

In this work, we performed controlled laboratory experiments to decipher the mechanistic relationship between soil moisture and microbial $H_2$ oxidation activity, focusing on conditions of negligible diffusive limitation. We analyzed three temperate soils and two arid soils from the Northeastern and Southwestern US, respectively. The temperate soils were sampled from three distinct ecosystems: a deciduous forest, a grass-shrub meadow, and an oak-pine forest. Across all soils, we measured $H_2$ oxidation rates in moisture-controlled incubations. Based on the previously established non-linear relationship between soil moisture and $H_2$ uptake, we expected the measured rates to be directly linked to the moisture response of HA-HOB, particularly under drier conditions when uptake is not diffusion limited. Thus, in incubations designed to minimize diffusive limitation for uptake, we determined the moisture sensitivity of HA-HOB by coupling measurements of water potential and microbial $H_2$ oxidation rates. We then revised previous formulations of soil $H_2$ uptake models based on the experimental data, and assessed the response of the global soil $H_2$ sink, both spatially and temporally, to the new parameterization. We further analyzed the microbial community structure in our temperate soil incubations to specifically examine how the diversity, abundance, and relative activity of different HA-hydrogenases (i.e., enzymes mediating high-affinity $H_2$ oxidation) and HA-HOB vary with moisture changes and soil type. Collectively, our findings provide mechanistic insights into how soil moisture regulates high-affinity $H_2$ uptake to improve our understanding of the global soil $H_2$ sink and its potential sensitivity to changing climate conditions.

## Results

### Moisture effects on microbial $H_2$ oxidation across soil types

Here we report microbial $H_2$ consumption measurements under controlled moisture conditions in three temperate soils—sandy loam and silty loam from a deciduous forest and meadow ecosystem, respectively, in central New Jersey, and loamy sand from the New Jersey Pine Barrens (PB). Hereafter, we refer to these as the forest, meadow, and PB soils, respectively. Following previous literature, we first report the results in terms of soil moisture. We then show that using the soil water potential rather than soil moisture offers better quantitative insights on the shared water-stress limitations for HA-HOB across soil types, as water potential provides a more intrinsic measure of water availability

that is comparable across different soil textures. Specifically for soil moisture, we express the experimental saturation as the fraction of water-filled soil pores, where 100% represents saturation and 0% corresponds to fully dried (oven-dried) conditions. However, since even after oven-drying, residual water remains trapped in smaller soil pores, we stress that 0% does not signify a complete absence of water—a known problem in the soil hydrology community[36]. As later discussed, this is an important caveat to consider, especially when evaluating the moisture percentages in the dry range.

Experiments revealed the expected non-linear relationship between soil moisture and $H_2$ oxidation activity in all three soils, with maximum uptake occurring below 40% saturation (Fig. 1a). $H_2$ consumption was not observed in control incubations (i.e., autoclaved soils, air-dried soils, and incubations without soils; Fig. S2). Measurable $H_2$ oxidation was observed under very low moisture levels, only slightly above air-dried conditions (Fig. 1a). Oxidation rates initially increased sharply with increasing moisture until reaching a maximum (Fig. 1a). Peak activity was observed at ~15%, 25% and 35% saturation in the PB, meadow and forest soils, respectively. Rates decreased above these moisture levels, indicating reduced microbial consumption of $H_2$ (Fig. 1a), and slight diffusive limitation in the PB soil above 60% saturation (Fig. S1). In PB sand, a clear decrease in uptake rate was not evident, as higher targeted saturation levels did not result in higher actual saturation, and uptake rates remained similar across moisture treatments. This pattern indicates drainage from the sandy soil, as expected at saturations above 30% (Fig. S1), impeding higher saturation levels in the incubations.

Across all three soils, the forest sandy loam soils showed the fastest microbial $H_2$ uptake with the highest measured rate constant (min$^{-1}$ g-drySoil$^{-1}$) 4- and 5-fold higher than that of the meadow silty loam and PB loamy sand soils, respectively. Uptake was not observed under fully air-dried conditions, which were estimated to correspond to approximately 5%, 7%, and 1.3% saturation for meadow, forest, and PB sand soils, respectively (Fig. S2, S3; Table S1). Measurable uptake was observed under moisture levels only slightly above air-dried conditions: ~5% saturation in silty loam (meadow), ~7% saturation in sandy loam (forest), and ~2% in PB sand (Fig. S3, Table S1). However, there is large uncertainty in these moisture measurements in the very dry range due to the small amount of water present. For example, for the forest soil, the lowest moisture amendment for which uptake was detected was estimated to have the same moisture content as the air-dried soil at ~7.6% saturation (Table S1). This highlights methodological limitations in estimating moisture content by conventional methods (i.e., oven-drying the soil sample until constant weight). Residual water that may still remain in the soil makes estimating moisture content gravimetrically in the dry end of the spectrum especially challenging. We therefore also measured the water potential ($\Psi$) of these soils as a more intrinsic measure of water availability.

### Unexpectedly low water stress threshold for high-affinity $H_2$ oxidizers

Water potential ($\Psi$), a measure of the energy state of water, is widely used as an indicator of soil water availability to plants and microbes (e.g.[37–39]). Our water potential measurements indeed showed that the air-dried samples had much lower water availability than those amended with moisture, even in the very dry range (Table S1). Moreover, the measurements revealed a low and consistent moisture condition for the onset of microbial $H_2$ consumption, which we define here as the water-stress threshold, $\Psi_{ws}$. In the forest soil, measurable $H_2$ oxidation occurred at $\Psi$ ~ -70 MPa, with no uptake in the air-dried soils at −130 to −140 MPa (Table S1). A similar $\Psi_{ws}$ was observed for the meadow soil, as the driest sample for which $H_2$ oxidation was observed was at ~65 MPa (Table S1). The PB sand soils were active under lower water potentials, down to approximately ~90 MPa (Table S1). We performed some additional tests on two sandy soils from arid regions.

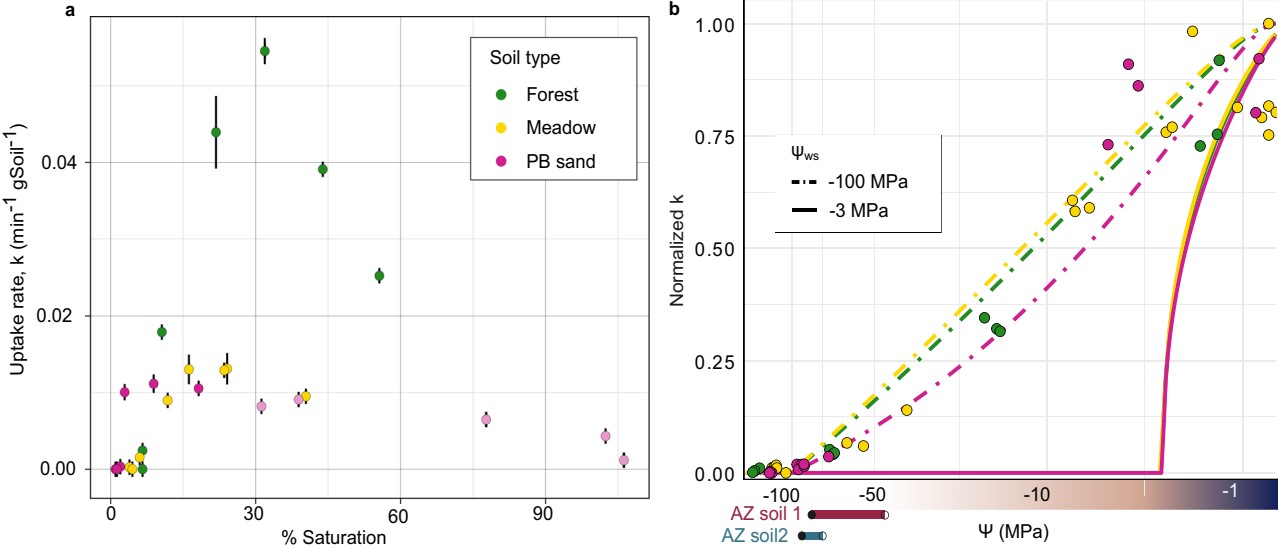

**Fig. 1 | Relationship between soil moisture and microbial H$_2$ oxidation.**
**a** Average H$_2$ oxidation rates (min$^{-1}$ gram dry-soil$^{-1}$) of triplicate incubations versus mean saturation (%). Error bars are standard errors around the mean ($n = 3$). Faded fill colors for PB sand indicate that the incubations did not show slower H$_2$ oxidation with rising moisture >30% saturation, likely owing to a difference between targeted and real saturation in the experimental setup (Fig. S1). **b** Average uptake rates (min$^{-1}$) normalized by maximum measured uptake rate in each soil type versus measured water potential $\Psi$ (MPa) of triplicate incubations. The curves are modeled uptake rates $f(s)*g(Ts)$ based on parameterizations developed in Ref. 26, revising the water-stress threshold ($\Psi_{ws}$ from -3 MPa to -100 MPa. The two shaded rectangles in (**b**) indicate the estimated $\Psi_{ws}$ for the AZ soils, taken as the range between the highest measured $\Psi$ with no H$_2$ oxidation (open symbol) and the lowest $\Psi$ (closed symbol) at which H$_2$ oxidation was observed (Table S2).

These desert soils exhibited significantly lower H$_2$ oxidation rates than the temperate soils in the moist regime, but their $\Psi_{ws}$ was similar to the temperate soils (non-zero oxidation rates measured for $\Psi > $ -75 MPa, Fig. 1b, Table S2). Collectively, these results suggest the water-stress threshold of HA-HOB to be between −70 and −100 MPa, among the lowest thresholds ever reported for microbial persistence and activity[37,38,40].

These measurements further allowed direct integration of our results into the recently developed mechanistic model for H$_2$ uptake that leverages $\Psi$ as a fundamental measure of moisture content[26] (Fig. 1b). Notably, this approach eliminates the need for soil-specific moisture thresholds for water stress and optimal conditions in parameterizing H$_2$ uptake as it directly links bacterial activity to thermodynamic considerations on water availability. Revising $\Psi_{ws}$ from −3 MPa, a value based on plant physiology used in the original model formulation[26], to −100 MPa, as measured here, yielded significantly improved fits to the experimental data (Fig. 1b).

## Uptake rates increase with soil carbon content

For the two loamy soils (forest and meadow), H$_2$ oxidation rates scaled with average soil organic carbon (SOC) content within the drier range of the moisture gradient (Fig. S4a). Peak oxidation rate in the forest soil was approximately 4-fold higher than that in the meadow soil on a per gram basis, when not controlled for SOC content. However, this difference disappeared when the rates were scaled by mean SOC (g/g), as the scaled rates were essentially equivalent for the two loam soils below the optimal moisture thresholds (i.e., <30% saturation). The loamy sand (PB sand), in contrast, showed higher rates than the two loams when adjusted for SOC content (Fig. S4a), suggesting higher relative abundance and/or activity of HA-HOB in these soils. Maximum H$_2$ oxidation rate ($k_{max}$ min$^{-1}$ g-Soil$^{-1}$) increased nearly linearly with SOC (Fig. S4b).

## Effects on the spatial and historical variability in H$_2$ sink strength

To examine how the observed sensitivities of microbial H$_2$ oxidation to moisture could impact the global H$_2$ soil sink, we estimate the deposition velocity v$_d$(H$_2$) (Eq. 3) using a $\Psi_{ws}$ of -3 MPa and -100 MPa. In this model, v$_d$(H$_2$) accounts for both biotic and physical processes regulating the soil sink, with simulations run with and without modulation by SOC (Fig. S4b). We find that the particular value of $\Psi_{ws}$ employed in the model significantly affects the spatial distribution of modeled uptake rates (Fig. 2a-c)[41]. For $\Psi_{ws} = −3$ MPa, soil moisture is predicted to be insufficient to support H$_2$ uptake in -90% of deserts and 45% of semi-arid regions. For $\Psi_{ws} = −100$ MPa, the degree of uptake inhibition is reduced to 70% (deserts) and 20% (semi-arid regions), respectively. In humid areas, where moisture is generally high enough to support microbial H$_2$ oxidation, v$_d$(H$_2$) is primarily controlled by diffusion[26].

A lower $\Psi_{ws}$ for HA-HOB tends to shift H$_2$ uptake from continental regions (-7 pp) to desert (+7 pp) and semi-arid (+4 pp) regions. The relative importance of these regions also impacts the predicted change in v$_d$(H$_2$) over the second half of the 20$^{th}$ century (Fig. 2d). In continental and polar regions, v$_d$(H$_2$) increase is insensitive to $\Psi_{ws}$, reflecting increased H$_2$ soil diffusion due to lower soil moisture, warming, and reduced snow cover[8,27]. In contrast, drying in desert and semi-arid regions has increased the areal extent of regions where soil moisture is too low to support H$_2$ uptake. As deserts play a larger role with a lower $\Psi_{ws}$, this contrasting response results in a smaller increase in v$_d$(H$_2$) (+7%) with a drier $\Psi_{ws}$ of -100 MPa, compared to a wetter $\Psi_{ws}$ of -3MPa (+-10.5%) from 1950 to 2019. Notably, a lower water-stress threshold of -100 MPa has a stronger effect on v$_d$(H$_2$) on interdecadal timescales compared to interannual timescales (Fig. S5).

We incorporated an SOC modulation into global simulations of v$_d$(H$_2$) using a linear fit given by $k_{max} = 0.00387 + 0.0024 * (SOC)$ to our data (Fig. S4b). Incorporating this modulation changes the relative contribution of continental regions, which are not moisture-limited, to H$_2$ uptake by nearly 35% (Fig. S6). The simulated relative contribution of hot deserts to the H$_2$ sink strength is also sensitive to SOC (Fig. 2; S6). Without SOC modulation (Fig. 2), the relative contribution of deserts to the overall H$_2$ sink increases by 75%. This SOC-induced change in the spatial distribution of the H$_2$ sink results in a weaker increase (20%) in the H$_2$ sink strength from 1950-2019.

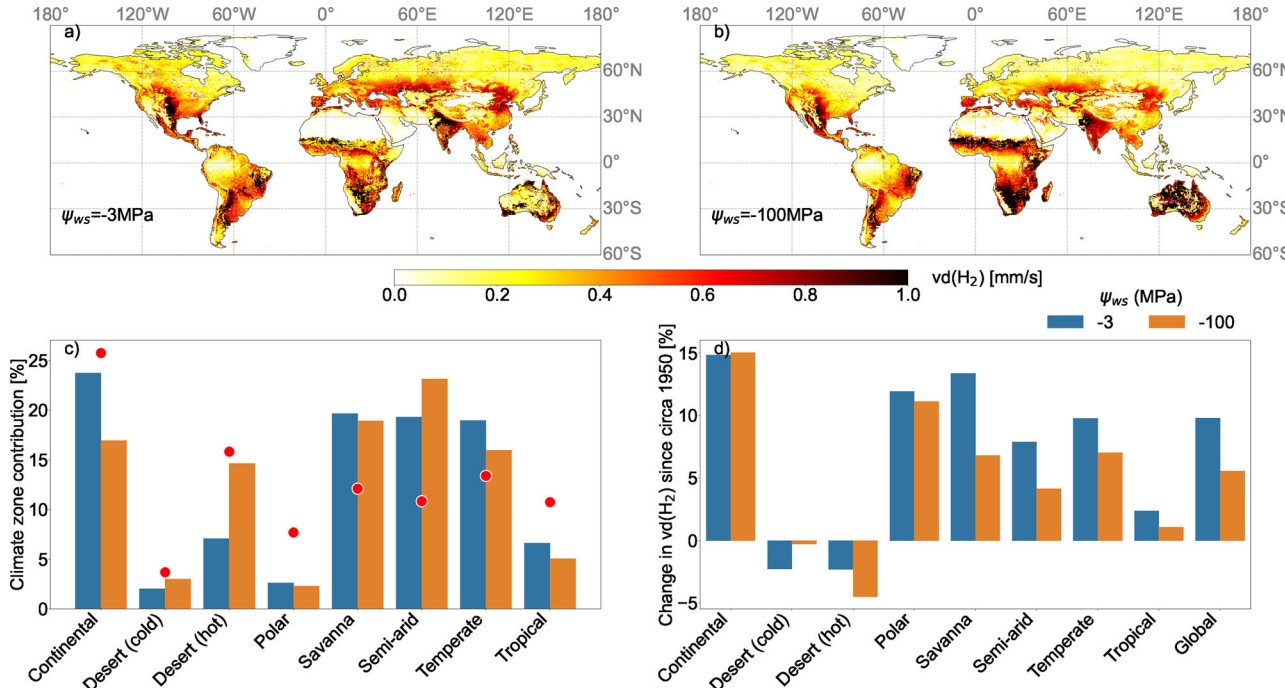

**Fig. 2 | Impact of a lower moisture threshold for activation on global H₂ uptake.** Spatial distribution of H₂ deposition velocity ($v_d(H_2)$ mm/s) for a water stress threshold of (**a**) -3 MPa vs (**b**) -100 MPa averaged from 2005 to 2019. $v_d(H_2)$ is prescribed to be 0.09 mm/s in both configurations for year 2010, as detailed in the Methods. Simulations without the SOC modulation are presented in Fig. S6. **c** Fractional contribution of different climate zones to $v_d(H_2)$ for different water stress thresholds. Red dots indicate the fractional landmass corresponding to each climate zone north of 60S. **d** Relative change in $v_d(H_2)$ across different climate zones from 1950–1964 to 2005–2019.

## Differential moisture response of high-affinity H₂ oxidizers

For each temperate soil type, we obtained metagenomes and metatranscriptomes at several moisture levels across the tested range (Table S4) to assess the community dynamics and ecophysiologies of high-affinity hydrogen oxidizers. Due to lower yield, replicate samples from PB sand and meadow soil were pooled before extraction. Putative high-affinity hydrogenase (HA-H₂ase) homologs were identified in both the contigs and metagenome-assembled genomes (MAGs) (see "Methods").

Across all three soils, group 1 h hydrogenase (Hhy) was the most dominant form of HA-H₂ases[15]; Table S4). Several Group 2a hydrogenases (with potential high-affinity activity[17,42]), were also recovered from the PB sand metagenomes, but these were significantly less abundant than the 1 h hydrogenases (3 homologs of group 2a versus 297 of group 1 h; Table S4). To estimate the relative abundance of HA-HOB in these soils, we compared the normalized abundances of HA-H₂ases with average normalized abundances of the single-copy marker genes for DNA gyrase, *gyrA* and *gyrB*. This analysis indicated that approximately 34% of the community in forest soil, ~25% in meadow soil, and 40% in PB sand have the ability for atmospheric H₂ oxidation (Fig. S7). These estimates are comparable to previous surveys of HA-HOB in various ecosystems, which indicated the relative abundance of hydrogen oxidizers to range from 20 to 57% of the total community across diverse ecosystems[43,44].

We next analyzed the relative expression (i.e., activity) of HA-H₂ases across moisture gradients in the three soils. For this analysis, we mapped metatranscriptomic reads to previously identified HA-H₂ase homologs. Mapped read counts were normalized by gene length and sequencing depth to obtain transcripts per million (TPM) values, providing a standardized measure of gene expression. In all three soils, active expression of HA-H₂ases was observed across all moisture levels (Fig. 3a). For the PB and Forest soils, TPM values did not differ significantly across moisture levels (Fig. 3a; Kruskal–Wallis test, p = 0.134

and 0.787 each), but differed significantly for the Meadow soil (p = 0.0004). We also assessed the combined total TPM of all HA-H₂ases at each moisture level, reflecting the community-wide magnitude of hydrogenase expression, and the median TPM across HA-H₂ases, which provides a robust measure of the typical expression level that is less influenced by highly expressed outliers. For the PB sand and forest soils, the total and median TPM values were relatively consistent across moisture levels (Fig. 3a-b), suggesting that HA-H₂ase transcription is broadly maintained across hydrological conditions in these soils. Total activity in the meadow soil generally increased with increasing moisture (Fig. 3c).

While the overall HA-H₂ase expression across moisture levels remained relatively constant in the forest and PB soils, there were notable differences in the specific hydrogenase genes being expressed at each moisture level across all three soils (Fig. 3d-f). In all three soils, specific moisture-adapted clusters of HA-H₂ases appeared to dominate at different moisture levels (Fig. 3d-f). This suggests moisture-related niche separation of HA-H₂ases, and locally contingent H₂ase populations. Furthermore, the lack of correspondence between total expression levels and measured HA-H₂ oxidation rates (Fig. 1a and Fig. 3a-c) suggest that the moisture-adapted clusters of HA-H₂ases likely differ in their catalytic efficiency.

High-quality MAGs encoding HA-H₂ases (all group 1 h) from the three soils were classified within the phyla Acidobacteriota, Actinomycetota, Eremiobacterota, and Chloroflexota (Fig. 4a). Despite recent evidence indicating atmospheric H₂ oxidation by Archaea[45], we did not find any archaeal MAGs with HA-H₂ase homologs. The specific genera within each phylum encoding HA-H₂ases differed across soil types (Fig. 4a). Relative abundances of several genera showed notable variations with moisture change. For example, in the forest soil, the Acidobacteriota genus UBA5704 was more abundant than all other HA-HOB genera under drier conditions, at ~7% saturation (Fig. 4a). The high-affinity hydrogenase in this MAG was also most highly expressed

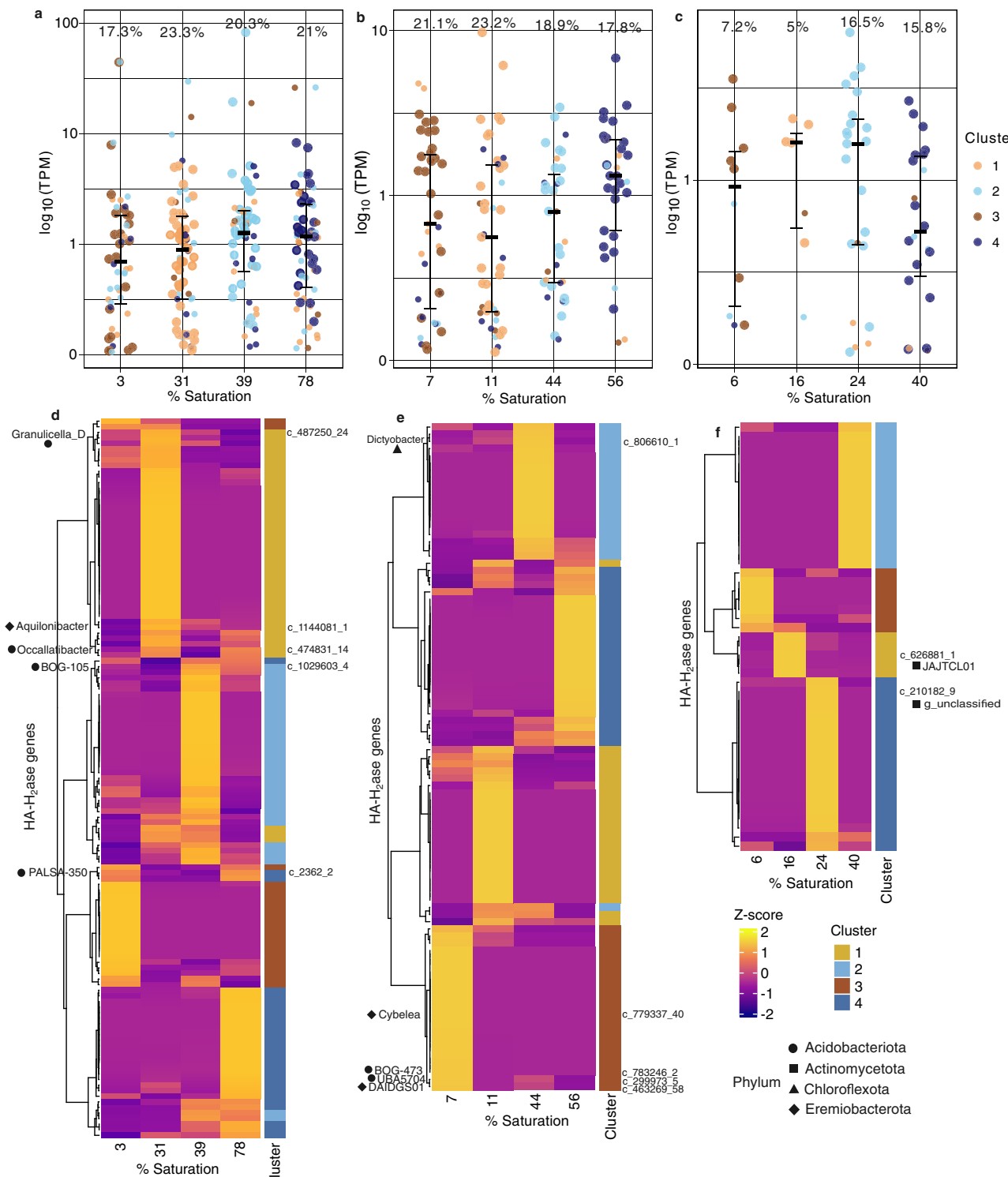

**Fig. 3 | Moisture-responsive expression patterns of HA-H$_2$ases across soil types.** **a–c** Distribution of normalized expression values (transcripts per million, TPM) for different HA-H$_2$ase gene clusters across moisture levels in (**a**) Pine Barrens loamy sand, (**b**) forest sandy loam, and (**c**) meadow silty loam soils. TPM values are log-transformed. Dominant H$_2$-Hase clusters for each moisture level are highlighted as larger points. Median TPM and error bars representing 25th and 75th percentiles are shown for each moisture level ($F$ = 129, 93, and 47 each for (**a**) PB sand, **b** forest, and **c** meadow soils, respectively). The numbers at the top of each panel indicate the percentage of active/expressed H$_2$-Hases among all identified HA-H$_2$ases. TPM values did not differ significantly across moisture levels for Pine Barrens (Kruskal-Wallis test, $n$ = 4, $p$ = 0.134) or Forest ($p$ = 0.787) soils, but differed significantly for Meadow soil ($p$ = 0.0004). Points are colored by their cluster IDs as identified in panels d-f. **d-f** Heatmaps showing relative expression of HA-H$_2$ases (rows) in PB sand (**d**), forest (**e**), and meadow soils (**f**). TPM values are clustered using k-means clustering (see "Methods" for details). Different gene clusters exhibit distinct moisture-responsive expression patterns. HA-H$_2$ases identified in metagenome-assembled genomes are indicated with gene IDs (where "c_" denotes the contig identifier), with the genus-level taxonomic assignments shown alongside the dendrogram. Symbols next to the genus names represent phylum-level classifications.

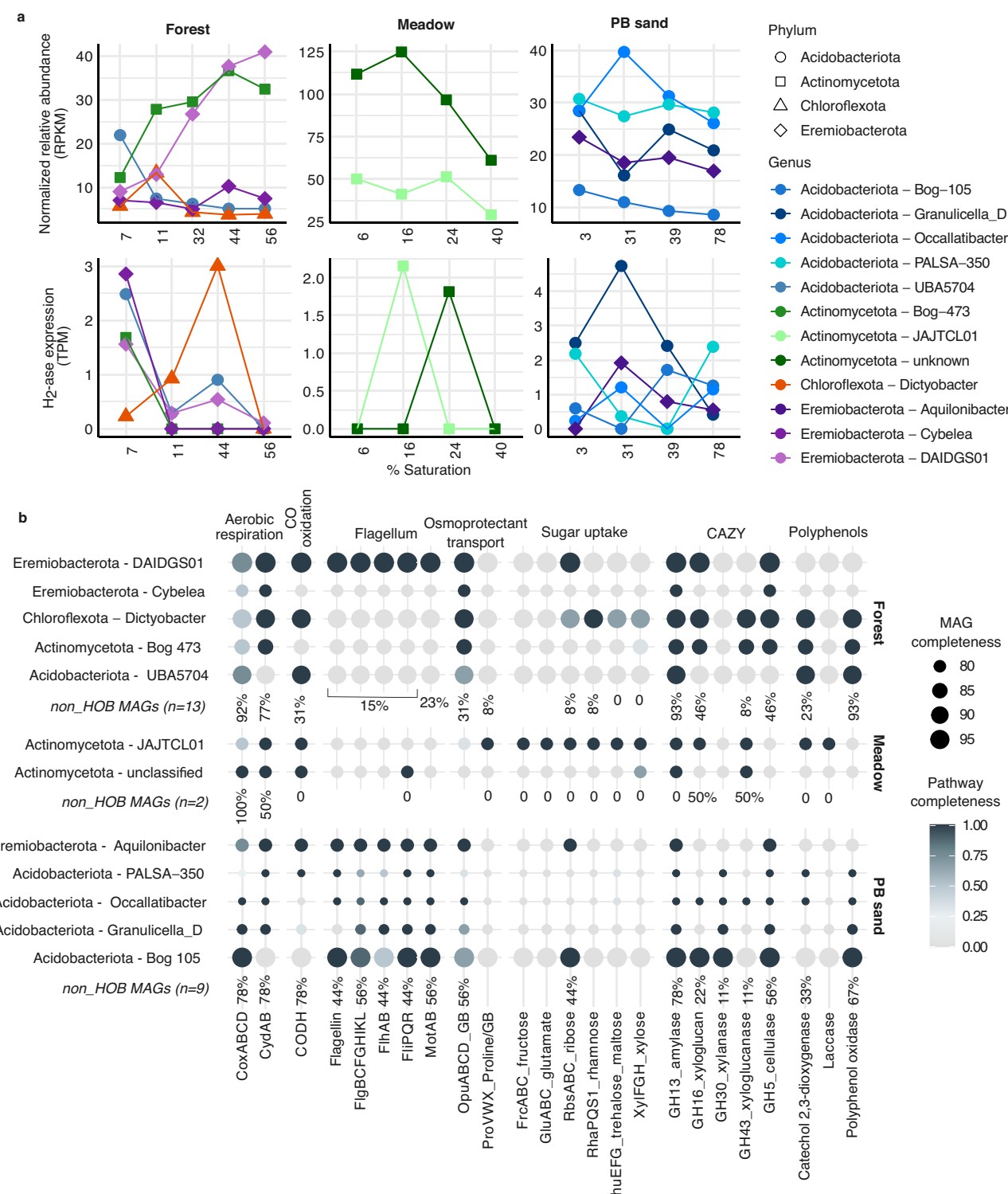

**Fig. 4 | Relative abundance, transcriptional activity, and metabolic potential of HA-HOB MAGs. a** Relative abundances and $H_2$-oxidation activity (top and bottom panels, respectively) of HA-hydrogenase-encoding MAGs across moisture levels in forest soil, meadow soil, and PB sand. Relative abundances in the metagenomes are expressed as number of reads mapped per kilobase of genome per megabase of metagenome (RPKM). Relative transcriptional activity of HA-$H_2$ases identified in each MAG is expressed as the number of reads mapped per million total reads

(transcripts per million, TPM). **b** Key metabolic pathways identified in the HOB MAGs. Pathway completeness is expressed as proportion of identified gene subunits corresponding to each pathway, and the subunits considered are indicated in the figure. For each soil, the proportion of non-HOB MAGs containing pathways detected in at least one HOB MAG is shown as a percentage. *CO* carbon monoxide, *GB* glycine betaine.

at 7% saturation (Fig. 4a). While the Eremiobacterota and Actinomycetota MAGs showed increasing relative abundances with increasing moisture in the forest soil, their HA-H$_2$ases were most highly expressed at the driest condition (Fig. 4a). These patterns collectively suggest their decreasing reliance on H$_2$ oxidation with increasing moisture.

In the meadow soil, the two Actinomycetota lineages appeared to mediate H$_2$ oxidation at two different moisture conditions (Fig. 4a). Similarly, the various lineages of Acidobacteriota and Eremiobacterota in the PB sand also showed diverging relative abundance patterns and variable expression of their HA-H$_2$ases with changing moisture. This further implies that different HA-HOB lineages mediate H$_2$ uptake across varying moisture levels in this soil type as well (Fig. 4a).

Functional profiling of the recovered HA-HOB MAGs revealed that aerobic heterotrophy was the predominant metabolic trait (Fig. 4b). While the capacity for the uptake of simple sugar compounds is relatively sparsely distributed, many of the MAGs harbor the potential for the degradation of complex carbon, including polyphenols and polysaccharides such as starch, xylan, and cellulose (Fig. 4b). These observations align with previous findings identifying HA-HOB as key players in the breakdown of complex plant-derived carbon in soils[46,47]. Flagellar motility is widespread among the PB sand MAGs (Fig. 4b)—likely advantageous in the highly varying moisture regimes characteristic of sandy soils with low water retention. Additionally, nearly all MAGs recovered from the PB sand and forest metagenomes also possess the ability for osmoprotectant transport, specifically for glycine betaine (Fig. 4b), which likely enables these HA-HOB to resist desiccation under dry conditions. H$_2$ oxidation likely provides an additional way to resist desiccation via the production of metabolic water, as has been hypothesized before[43,48]. Notably, many of the MAGs also possess the ability for carbon monoxide oxidation (Fig. 4b), highlighting their capacity for versatile trace gas metabolism. Overall, functional profiling of the HA-HOB MAGs suggests that they possess multiple metabolic and physiological strategies—including the ability to oxidize atmospheric H$_2$—that likely enables persistence and survival under challenging environmental conditions.

## Discussion

The H$_2$ soil sink is characterized by a water-stress or activation threshold for HA-HOB activity, below which there is no H$_2$ uptake. To account for the variability in moisture content with soil texture, soil water potential has been suggested as a more accurate representation of the true water availability for hydrogen oxidizers[26]. A value of -3 MPa was originally suggested, based on an analogy with the typical water-stress threshold for plant wilting point in semi-arid regions[26,49]. Following the discovery that HA-HOB are widespread in arid environments such as Antarctic desert soils[19], recent model implementations have used an even lower threshold of −10 MPa[27]. The specific parameterization of the moisture threshold substantially affects model outputs, which makes it challenging to extrapolate spatial and temporal variability in uptake and attribute recent increases in atmospheric H$_2$ levels to specific drivers[27].

Our results indicate a substantially lower moisture threshold for microbial H$_2$ oxidation (closer to −100 MPa; Table S1 and Fig. 1). These are some of the lowest water potential values reported for microbial activity, and life in general, in soils, comparable only to previous observations of CO oxidation in saline volcanic ash soils (down to −117 MPa[38,40]). Moisture thresholds that have been proposed and experimentally determined for other microbial processes are generally much higher. Previous studies have identified -14 MPa as the threshold below which microbial respiration ceases[37,50]. For nitrification, this threshold has been identified as -6 MPa[51]. While the widespread occurrence of HA-HOB in hyper-arid cold (e.g., Antarctic) and hot (e.g., Atacama) deserts[19,48] suggests a low water stress threshold for these microbes, our results provide quantitative constraints on the moisture limits of HA-HOB activity. The threshold values determined here are thus

critical for developing a mechanistic understanding of the ecophysiology of HA-HOB, particularly in the face of global change. Importantly, the similarity between the water-stress thresholds for uptake across all five soil types (forest sandy loam, meadow silty loam, PB loamy sand, and two arid soils—a loamy sand and sandy loam; Tables S1 and S2) is promising for global extrapolation of these results when modeling the soil H$_2$ sink. With a lower threshold for activation, the importance of arid regions in H$_2$ uptake increases significantly, which is noteworthy given the projected expansion of drylands with climate change[52,53].

Other key aspects of current global model parameterizations that require careful assessment include assumptions that HA-HOB are equally distributed across soil types[8]. Similar to the results of numerous previous microbial surveys in diverse ecosystems, our microbial analyses suggest that this assumption is an oversimplification (Figs. 3, 4). The relative abundances and activity of HA-HOB vary with soil and ecosystem type, aridity gradients, and SOC content (Figs. 3, 4, S7; e.g.[18,19,44,48,54–56]). Thus, integrating a more accurate representation of their population dynamics and ecophysiologies in global H$_2$ models may help explain the difference in predicted activity between deserts versus temperate ecosystems[57]. Our results show a positive correlation between SOC and H$_2$ activity (Fig. S4), aligning with previous observations[58]. Integrating this SOC scaling or even more complex parameterizations could further reduce uncertainties in soil H$_2$ oxidation potential. However, the overall impact of this modulation on H$_2$ deposition velocity (the flux as seen from the atmosphere) remains relatively minor compared to the dominant influence of soil moisture (Fig. 2 and Fig. S6). Indeed, while SOC regulates the biotic potential for hydrogen consumption, affecting the relative importance of different regions for H$_2$ uptake, soil moisture influences both microbial activity and the physical diffusion of H$_2$ through the soil matrix to remain the major modulator of H$_2$ soil uptake. Better constraining the mechanistic links between SOC and HA-HOB distribution and activity will be crucial to improving predictions of spatial distribution of the soil H$_2$ sink strength under global change.

From a microbial point of view, carbon-rich soils (e.g., forest soil) may exhibit more intense resource competition, leading to higher reliance of the microbial community on alternative energy sources such as H$_2$. A majority of the forest soil HA-HOB MAGs also appear to be specializing in complex carbon degradation, suggesting their occupance of an oligotrophic niche, even in carbon-rich soils (Fig. 4b). Their mixotrophic lifestyle implies that in such soils, heightened competition for carbon substrates due to higher microbial biomass could enhance mixotrophic activity, including H$_2$ oxidation. In more oligotrophic ecosystems such as desert soils, local gradients in SOC may select for oligotrophic H$_2$ oxidizers due to their ability to withstand carbon-poor conditions[56]. However, the lower microbial biomass in such soils still results in lower net uptake rates compared to more carbon-rich soils.

In general, the specific reasons for microbial H$_2$ oxidation remain unclear, even though various lines of evidence point to this being a survival strategy under resource limitation[43]. On the drier end of the moisture spectrum, it has been proposed that H$_2$ oxidation may provide metabolic water that could enhance survival[48]. Yet water addition has also been demonstrated to substantially stimulate H$_2$ oxidation activity in desert soils[19], highlighting the role of H$_2$ as an energy source for microbes in oligotrophic systems. Our genome-centric analyses reveal that HA-HOB are likely able to withstand desiccation by using osmoprotectants (Fig. 4b), further explaining their activity in extremely dry soils. Further studies are required to decipher the complex mechanisms regulating the ecophysiological response of HA-HOB to moisture variability.

Soil moisture and its sensitivity to environmental changes remain challenging to represent in global models[59], producing conflicting projections regarding future changes in aridity[60]. With a lower

moisture threshold that expands the importance of arid lands for $H_2$ uptake, reducing uncertainty in future aridity changes will be important for global $H_2$ consumption projections. In this regard, another important variable to consider is the change in SOC. Warming is expected to reduce average soil carbon globally[61,62]—any effect this will have on $v_d(H_2)$ remains unclear due to unknown mechanisms governing HA-HOB activity and SOC variability, both in abundance and form. Better constraining the mechanistic links between SOC and HA-HOB distribution and activity will be crucial to improving predictions of spatial distribution of the soil $H_2$ sink strength under global change. Such knowledge is also critical to advance historical reconstructions of atmospheric $H_2$[41], clarifying long-term trends and variations in the global $H_2$ budget. Finally, given the variety of soil biogeophysical conditions worldwide, applying our approaches to a broader range of soil types will provide additional evidence to the observed patterns.

## Methods

### Soil collection, characterization, and experimental setup

The temperate soils were collected from three distinct ecosystems: a deciduous forest, a managed grassland shrub meadow, and a sandy oak-pine forest. The forest and meadow ecosystems are located within the Watershed Research Institute in Pennington, New Jersey ($40.35501^0$, $-74.77458^0$, and $40.35360^0$, $-74.77433^0$, respectively). The oak-pine forest is located within the New Jersey Pinelands/Pine Barrens, and we sampled in the Brendan T. Byrne State Forest ($39.88683^0$, $-74.53663^0$). The arid soils were collected from two locations in Arizona (AZ soil 1: $34.73670^0$, $−111.98541^0$; AZ soil 2: $33.96932^0$, $−112.12924^0$). At each location, soil samples ( <1 kg) were collected from three sub-locations spaced approximately 1 m apart, sampling within the top ~10 cm, after removing the living organic layer, if present. Subsamples from each location were combined into a composite sample by mixing in pre-cleaned (acid-washed, UV-sterilized) sealable plastic bags and stored on ice while transporting back to the laboratory. At the lab, soils were spread into a thin layer on a clean tray lined with an acid-washed and UV-sterilized plastic bag, any large plant material was manually removed, and the samples were left to air-dry for up to 3 days (until visually dry). The air-dried soils were sieved through a clean 2 mm sieve and stored in pre-cleaned sealable polyethylene bags in the refrigerator. All moisture incubations were set up within 10–15 days of sieving and storage.

Before setting up the moisture incubations, the sieved soils were characterized for soil texture, pH, porosity, bulk density, and gravimetric moisture content. Soil texture analysis revealed the forest soil to be sandy loam, the meadow soil to be silty loam, and the Pine Barrens soil to be loamy sand. The two arid soil textures were determined to be loamy sand and sandy loam. All three temperate soils were acidic (forest soil pH 4.2, meadow soil pH 4.6, PB sand pH 4.1). Measured porosity values for the sieved soils ranged from ~0.6 for both the meadow and forest soils, ~0.4 for PB sand, and ~0.5 for the two arid soils. Gravimetric moisture content (i.e., g water/g dry-soil) was determined by oven-drying soil at 105 $^0$C until constant weight. Volumetric moisture content was then estimated by multiplying gravimetric moisture content by bulk density.

Soil organic matter content was estimated using the loss on ignition (LOI) method. Briefly, triplicate air-dried soil samples were oven-dried at 105 °C until constant weight to remove moisture. The dry weight was recorded, after which samples were combusted in a muffle furnace at 505 °C for 4 h to oxidize organic matter. LOI (%) was calculated as:

$$LOI\ (\%) = \left( \frac{Initial\ dry\ weight - Combusted\ weight}{Initial\ dry\ weight} \right) \times 100 \quad (1)$$

LOI values were converted to soil organic carbon as[63,64]:

$$SOC\ (\%) = 0.58 \times LOI(\%) \quad (2)$$

To set up the moisture-amended incubations, we added a small amount of air-dried soil (2 g each of forest, meadow, and arid soils, and 4 g of PB sand) into acid-washed and autoclaved 240 ml serum bottles. We then slowly added ultrapure water to the soil to achieve specific moisture levels (3 replicates per moisture level), mixed by gently shaking the bottle. A sterile autoclaved polypropylene spatula was occasionally used to gently mix the soil to facilitate water distribution. The bottle was quickly sealed with parafilm to reduce evaporative losses, wrapped in aluminum foil, and incubated at 22 $^0$C. The acclimation period lasted ~6 days (i.e., all measurements were taken within 5–7 days of incubation) for the temperate soils. Any evaporative water loss during the incubation period was assessed by weighing the bottles daily. No measurable weight loss occurred during the incubation period. We further determined that the experiments were not diffusion-limited in the dry range (Fig. S1), as described in Supplementary Methods. The arid soil incubations were acclimated overnight before $H_2$ oxidation measurements.

### $H_2$ oxidation measurements and soil moisture characterization

For $H_2$ oxidation rate measurements, incubation bottles were retrieved from the incubator, flushed with room air, and immediately closed with a butyl rubber stopper, pre-boiled in sodium hydroxide, and crimped with aluminum seal. Experiments were conducted at ambient $H_2$ concentrations at 22 $^0$C. Time zero sample (2 mL) was immediately drawn from the bottle headspace using a luer-lock syringe and injected into a reducing compound photometer (RCP) gas chromatograph (Peak Laboratories LLC) fitted with a 1 mL sample loop. Additional headspace samples (2 mL) were collected and immediately measured on the GC-RCP until an exponentially decreasing trend in $H_2$ concentrations was captured. Triplicate incubations for each soil type-moisture combination were measured in parallel, with headspace samples sequentially drawn and analyzed from each replicate. Most experiments lasted under 2 h. The AZ soils were much slower in consuming $H_2$, and therefore, the experiments ran substantially longer ( ~9 hr for AZ soil 1, ~50 hr for AZ soil 2). The limit of detection for the measurements was determined to be 57 ppb. Accordingly, the limits of detection for $H_2$ oxidation rates were approximately 0.1 ppb min$^{-1}$ g-Soil$^{-1}$ (for incubations containing 2 g soil; forest and meadow soils), 0.04 ppb min$^{-1}$ g-Soil$^{-1}$ (for PB soil incubations containing 4 g soil), 0.05 ppb min$^{-1}$ g-Soil$^{-1}$ for AZ soil 1, and 0.01 ppb min$^{-1}$ g-Soil$^{-1}$ for AZ soil 2. Headspace $H_2$ drawdown was modeled as a first-order decay process, and the rate constants (min$^{-1}$) were used as a proxy for the rate of $H_2$ oxidation. Initial $H_2$ concentrations matched the corresponding ambient $H_2$ levels measured for each experiment. For each soil type, killed controls (i.e., containing autoclaved airdried soil) and no-soil (empty) controls were also measured for $H_2$ uptake (Fig. S2).

Immediately following uptake measurements, the bottles were briefly sealed with parafilm again and re-weighed to determine evaporative losses during measurement. We did not observe any measurable weight loss during measurements. Soil was quickly subsampled into pre-weighed receptacles for: (i) re-measuring gravimetric water content (0.5 to 1 g soil), (ii) water potential measurements, and (iii) DNA/RNA. Approximately 1-2 ml of soil ( ~0.5–1 g) from the incubations was used for water potential measurements using the WP4C dewpoint potentiometer (METER Group). Soil subsamples for DNA/RNA ( ~0.3-0.5 g each per replicate) were sampled into sterile cryovials and frozen at -80 C promptly following sampling.

### Nucleic acid extractions and sequencing

Microbial community analysis —including nucleic acid extraction, sequencing, and bioinformatic processing—was conducted for the

temperate soils using widely established and standard protocols as described below. Total nucleic acids (DNA and RNA) were co-extracted (in two separate fractions) by combining the RNeasy PowerSoil Total RNA kit with the RNeasy PowerSoil DNA Elution kit (Qiagen), using manufacturer's protocols. Quality and yield of the extracted DNA and RNA samples were assessed using NanoDrop and Qubit fluorometer respectively. For the forest soil, replicate soil samples from each moisture perturbation (~ 0.3-0.4 g) were extracted separately as these samples generally yielded high DNA/RNA concentrations (Table S3). In contrast, for the meadow soil and PB sand, triplicate soil samples per moisture amendment were pooled prior to extraction (~ 1.2 g for pooled soils; Table S3). DNA extracts were cleaned using the QIAGEN DNeasy PowerClean Pro Cleanup kit to improve quality and were re-measured on the NanoDrop to ensure minimal impurity levels in the final extracts. Extracted nucleic acids were stored at -80 °C until further processing.

DNA and RNA samples corresponding to 4–5 moisture levels for each soil type were used for metagenome and metatranscriptome sequencing (Table S4). Total DNA and RNA samples for the selected samples were sent to the Princeton Genomic Core facility for paired-end sequencing on the Illumina NovaSeq platform (S1 300nt flowcell). For the RNA samples, a ribo-depletion step was performed prior to library construction. The resulting reads were adapter-trimmed and demultiplexed by the sequencing facility. Further processing of the reads was carried out using the Princeton Research Computing cluster.

### Processing and analyses of metagenomes and metatranscriptomes

Demultiplexed metagenomic reads were quality filtered using FastQC (v0.11.9)[65] and Trimmomatic (v0.39)[66]. Metagenomes for each soil type were co-assembled using MEGAHIT (--min-contig-len 1000; v1.2.9)[67,68]. Bowtie 2 (v2.3.5)[69] was used to map reads back to the assembled contigs. Contig gene-calling was performed by using Prodigal (v2.6.3)[70] and the resulting gene sequences were used as input for DIAMOND blastp (v2.1.10)[71] search against the UNIPROT-SWISSPROT database to obtain functional annotations. Contigs longer than 1500 bp were binned using MetaBAT 2 (v1.12.1)[72] and MaxBin2 (v2.2.7)[73]. Bin refinement was carried out using the bin_refinement module in MetaWRAP (v1.2)[74]. Prodigal (v2.6.3)[70] was used to obtain amino acid sequences of gene calls for each bin. CheckM (v1.2.3)[75] was used for bin quality assessment. We only retained medium- to high-quality (> 70% complete and <10% contamination) genomes for downstream analyses. Taxonomic annotations were obtained by using the Genome Taxonomy Database toolkit (v.2.4.0)[76,77], classifying against the R220 database.

Metatranscriptome reads were QC-filtered to remove adapters by using FastQC (v0.11.9)[65] and Cutadapt (v3.7)[78]. Using Bowtie 2 (v2.3.5)[69], QC-filtered reads were mapped to the metagenomic contigs obtained for each soil type. Relative abundance estimates for contigs and metagenome-assembled genomes (MAGs) were obtained by using CoverM (v0.7.0)[79]. For abundance estimates of MAGs in the metagenomes, Coverm "--method rpkm" was used. Conversely, "--method tpm" was used for estimating relative transcription of the MAGs in the metatranscriptomes. Additional functional annotations for MAGs were obtained via DRAM (v0.1.2)[80] and RASTtk (v1.073)[81] as implemented within the KBase platform[82]. KO annotations for translated proteins in each MAG were obtained using GhostKOALA[83]. DIAMOND BLASTp[71] searches against databases generated for each hydrogenase group from HydDB[84] were used identify putative hydrogenase homologs in MAGs and contigs. Hits were aligned with reference sequences using Mafft (v7.475)[85], which were filtered using trimAl (v1.4.1; "-gt 0.8 -cons 60")[86], then used to compute phylogenetic trees using FastTreeMP (v2.1.10)[87]. Potential false positives were filtered out based on phylogenetic placement, homology, and residue comparisons.

Metatranscriptomic reads were mapped to the high-affinity hydrogenase homologs identified in the metagenomic contigs.

Mapping was performed using Bowtie 2 (v2.3.5)[69] and Samtools (v1.9)[88]. Mapped read counts were normalized by gene length and total metatranscriptome read counts to obtain transcripts per million (TPM) values for each gene homolog. Relative expression of high-affinity hydrogenases in the three soils was compared across moisture levels by using heatmaps (ComplexHeatmap package in R) with k-means clustering. To determine an appropriate number of clusters (k), we applied the silhouette method using the fviz_nbclust() function from the factoextra R package, which evaluates clustering performance across different k values based on average silhouette width. Based on this analysis, k = 4 was selected as the optimal number of clusters. K-means clustering was then performed on z-score–normalized TPM values using the kmeans() function in R with 25 random starts to ensure stability of the clustering solution. Statistical differences in TPM across moisture levels within each soil type were assessed using the Kruskal–Wallis test with $p$-values adjusted for multiple comparisons.

### Global estimates of $H_2$ deposition velocity, $v_d(H_2)$

The deposition velocity of $H_2$ ($v_d(H_2)$) is calculated with the solution of the vertically explicit $H_2$ mass balance in soil[89,90] as:

$$\frac{1}{v_d(H_2)} = \frac{1}{g_i} + \frac{1}{g_s} \tag{3}$$

where $g_i$ represents the $H_2$ conductance through possible diffusive barriers (e.g., snow cover, canopy), which reduces $H_2$ transport to HA-HOB, and $g_s$ is the conductance through the active soil layer where $H_2$ oxidation by HA-HOB occurs. $v_d(H_2)$ in this model accounts for both biotic and physical processes regulating the soil sink. $g_s$ can be expressed as:

$$g_s = \sqrt{k_m f(s) g(T_s) h(SOC) D_s} \tag{4}$$

where $f(s)$, $g(Ts)$[23], and $h(SOC)$ are the sensitivity of microbial $H_2$ oxidation to volumetric soil moisture (s), temperature (Ts), and SOC, respectively[26]. $D_s$ is the diffusion of $H_2$ in the soil and $k_m$ is the maximum oxidation rate of $H_2$ by HA-HOB. $f(s)$ is parameterized following Bertagni (2021)[26] as:

$$f(s) = \frac{1}{N} \left(s - s_{ws}\right)^{\beta_1} \left(s_{opt} - s\right)^{\beta_2} \tag{5}$$

where $s_{ws}$ is the lower moisture threshold for bacterial activity, and $s_{opt}$ is the optimum. These soil-specific moisture values can be derived using only two water potential values (for bacterial inhibition and optimum) and the soil water characteristic curves. For our simulations, these water potential values are derived from the experiments. N is a normalization factor such that $\max(f) = 1*h(SOC)$ was empirically derived based on the experiments performed in this study as a linear fit between maximum of $H_2$ oxidation rate for each soil type and mean SOC (%) (described in Results).

$$\frac{1}{g_i} = \frac{1}{g_{snow}} + \frac{1}{g_{cano}} \tag{6}$$

Term $g_i$ reflects the transport of $H_2$ through the canopy ($g_{cano}$) and through snow ($g_{snow}$); $g_{snow}$ was calculated from snow depth assuming a snow porosity of 0.645; $g_{cano}$ was prescribed following[91]. Given the importance of short-term hydrological fluctuations[26], we used hourly soil moisture, temperature, and pressure from ERA5 reanalysis[92,93]. The time-invariant spatial distribution of SOC was prescribed following Soil Grids v2.0 (top 5 cm[94]).

$k_m$ was optimized for each configuration, such that the global deposition average $v_d(H_2)$ is 0.09 mm/s for year 2010[8].

All statistical analyses and plotting were done in R (v4.4.1[95]) and Python.

## Reporting summary

Further information on research design is available in the Nature Portfolio Reporting Summary linked to this article.

## Data availability

The data that support the findings of this study are available at the GitHub repository: (https://github.com/Linta-Reji/Reji2025_H2_SoilMoisture). Measured $H_2$ oxidation rates and soil properties have been uploaded to FigShare (https://doi.org/10.6084/m9.figshare.28652099). Metagenomes and metatranscriptomes generated in this study have been deposited in the NCBI Sequence Read Archive, BioProject ID PRJNA1330882.

## Code availability

The code used in this study is available at (https://github.com/Linta-Reji/Reji2025_H2_SoilMoisture[96]).

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

## Acknowledgements

This research has been supported by the Water Grand Challenge Award from the High Meadows Environmental Institute at Princeton University (X.Z., L.R.), the Carbon Mitigation Initiative (CMI) at Princeton University (X.Z.), and by the US Department of Energy, Office of Energy Efficiency and Renewable Energy (EERE), specifically the Hydrogen and Fuel Cell Technologies Office (award NA23OAR4310138, X.Z.). The views expressed herein do not necessarily represent the views of NOAA, the U.S. Department of Energy, or the United States Government. This material is based upon work supported by the High Meadows Environmental Institute at Princeton University. We thank the Watershed Institute (NJ) and the New Jersey Conservation Foundation for help with sample acquisition. We thank Dr. Amilcare Porporato and Damola Olaitan at the Department of Civil and Environmental Engineering at Princeton University for their valuable feedback during the execution of the project. The work reported on in this paper was substantially performed using the Princeton Research Computing resources at Princeton University which is a consortium of groups led by the Princeton Institute for Computational Science and Engineering (PICSciE) and Office of Information Technology's Research Computing. This work was supported by the Office of Science, Office of Biological and Environmental Research, of the US Department of Energy under Award Numbers DE-AC02-05CH11231, DE-AC02- 06CH11357, DE-AC05-00OR22725, and DE-AC02-98CH10886, as part of the DOE Systems Biology Knowledgebase.

## Author contributions

L.R., X.Z., M.B., and F.P. designed experiments, L.R. and Q.Q. performed uptake experiments and analyzed the data. L.R. collected and analyzed omics data. M.B., F. P., and L.R. performed modeling. X.Z. provided resources, funding, and supervision for the project. L.R. wrote the initial draft of the manuscript. All authors contributed to data interpretation, manuscript writing, and approved the final version.

## Competing interests

The authors declare no competing interests.
