## [Transparent Peer Review file · Nature Communications]

Global Implications of a Low Soil Moisture Threshold for Microbial Hydrogen Uptake

Corresponding Author: Professor Linta Reji

Version 0:

Reviewer comments:

Reviewer #1

(Remarks to the Author)

Dear Authors and Editors,

Reji, Bertagni, Paulot, Qin and Zhang present new measurements combined with analysis that offers a wealth of new insight into soil microbial uptake of hydrogen. The authors correctly identify that this soil sink remains poorly constrained. They conduct new measurements that inform how the soil sink is related to the availability of soil water to microbes. The analysis then identifies contenders for hydrogen oxidising microbes present in the tested soils and, by inserting their water potential constraint into a theoretical soil uptake model, probes the planetary distribution of soil uptake by climate zones.

I consider that this work is a useful contribution to our understanding of the hydrogen soil uptake in low-moisture soils. The particular significance of their results comes from their finding that hydrogen oxidising microbes uptake hydrogen even when water is much harder to obtain from soils than the wilting point of plants. I expect that there is wide interest in how this underscores the fascinating nature of hydrogen as an important energy source for microbes in the most challenging environments. For those of us focused on the hydrogen budget, this work has important implications for how soil uptake may be included in atmospheric chemistry simulations – the authors find a more important role for uptake in arid regions – and may carry implications for policy-important hydrogen GWP100 calculations.

One disagreement between this study and an ongoing hydrogen uptake study (Drewer, Cowan, Nemitz et al., shared publicly: <https://h2envimpacts.org.uk/environmental-impacts-of-hydrogen-energy-meeting/> : <https://drive.google.com/drive/folders/1Dvm5pzbnL8D-mA3EMzFTqnEVAKKYUKB>) is in their choice of just several grammes of soil (ln 180-210) compared with '~800 g', 'packed to roughly field bulk density', in the Drewer et al. method. Please can the authors explain why they choose such a small mass. Additionally, please can they comment on the implications for diffusive or soil-moisture dynamics of having such a relatively large surface-area to volume in these experiments compared to others.

Can the authors include an explicit comparison between their results and with the recent work of Paulot et al. (ACP,2024), where similar schemes with $\Psi_{ws} = -3$ MPa and -10 MPa are compared using GLDAS data instead of ERA5 hourly (ln 300)?

An additional analysis that would boost the importance of this work is a quantification of the sub-decadal variability in soil uptake with water-stress at -100 MPa compared with water-stress at -3 MPa. This is a small extension from their calculated 1950-2019 uptake change (ln 438-441) that would provide useful insights into a poorly understood variability of atmospheric hydrogen.

Signed cordially,

Alex Tardito Chaudhri

Minor comments:

(ln 39) suggestion: these water potentials could be contextualised, e.g. ln 354,535 -3 MPa is mentioned as the wilting point

for semi-arid plants.

(In 69-70) different estimates vary from 60s%-80s% so I'm not convinced that we can confidently state that soil uptake is certainly over 70% of sink. E.g., based on atmospheric hydrogen's isotopic signature Rhee et al. 2006 get over 80%, but based on some assumptions and what works in their models the Sand et al. 2023 estimate c.70% but with some models less than that.

(In 89-90) this mention of the 'paucity of observational constraints' is very important, and I am impressed that the authors seek to address this. Suggestion: however, there is a relative abundance of measurements in temperate NH America, so is it worth highlighting the particular paucity of soil flux measurements in (semi-)arid soils?

(In 126,127,174) there is a lack of consistency in lat-lon reporting. For usability, I would suggest reporting in degrees with decimals in both cases, and for decimal places is it important or accurate that a location is reported to a precision equivalent to one metre?

(In 126,127) reliability: are there issues with collecting samples in single locations in your sample areas? I understand that in-situ flux chamber measurements show enormous spatial variability uptake within metres of each other (e.g. Cowen et al. EGU sphere preprint-2024).

(In 154) hyphen formatted as minus sign.

(In 196-198) suggest these rates would be clearer to understand in ppb/min/g (to check – sometimes units formatted gSoil, other times as g-Soil).

(In 198-200) is this as trivial as discussed here? Chamber uptake of hydrogen is seen to occur on very short timescales, please explain how this model of drawdown is robust.

(In 215-266) I indicate that these sections are beyond my expertise. Suggestion: is it worth commenting how widely established these methods to give context to readers without this expertise?

(In 317-318) suggestion: considering your helpful phrasing about the intrinsic nature of Ψ (In 354-355,379-381), could the link be better explained here? Also, do your ideas in In 550-553 not give some good justification for why you might want to make this link sooner?

(Fig. 1) are the scattered points the same tests in each panel, can the triangles, circles, squares in a also be used in b. The legend in b shouldn't have the scatter marker lying on a line.

(In 434-436) please see commentary above regarding quantifying sub-decadal variability of uptake.

(In 477) typo 'to be range from'

(In 484) indicating

(In 498-499) suggestion: is it worth commenting on how usual/unusual it is to find a novel genus in such tests, e.g. is it convincing that it is a novel genus rather than a measurement error?

(Fig. 3) Clarity: as each genus can only be in one phylum please can the legend be simplified to list the genera under sub-headings for the phyla with the correct symbols (currently they all look like acidobacteriota in the legend and it is hard to determine colours). Then can you omit the 'p__'s and 'g__'s? Is 'Flgellum' a misspelling and should 'flagellin' have a capital F?

(In 576-578) this might be true in the sub-tropics and tropics, and presumably also when you integrate everywhere, but can be potentially misleading if soil moisture is not the primary driver everywhere. In (23) and (26), $h(T)$ has a steep gradient between 0 and 20 degC. (35) see that the observed hydrogen distribution can be explained where NH extra-tropics mostly respond to temperature seasonality.

(Remarks on code availability)

https://github.com/Linta-Reji/Reji2025_H2_SoilMoisture provides clear instructions and data files are available. (I have searched through the repository but not have not run the code in R).

Reviewer #2

(Remarks to the Author)

Journal-specific considerations

- What are the noteworthy results?

This article is the first to systematically relate soil H₂ uptake to water potential, and the authors show that H₂ oxidation by microbes occurs at some of the lowest water potentials of any known microbial process. The authors implement new

understanding of the relationship of H₂ uptake and soil water-potential into a global model, which shows that arid lands likely make a relatively larger contribution to the global H₂ sink

- Will the work be of significance to the field and related fields? How does it compare to the established literature? If the work is not original, please provide relevant references.

Yes, this work represents a significant advancement to our understanding of the moisture sensitivity of overwhelming soil sink for atmospheric H₂. The field is limited in data, and this systematic study that was well-designed to meet the needs for upscaling with models makes an important contribution to our understanding of its major driver of variability.

- Does the work support the conclusions and claims, or is additional evidence needed?

Yes.

- Are there any flaws in the data analysis, interpretation and conclusions? - Do these prohibit publication or require revision?

We suggest that the authors focus their analysis of metagenomes and metatranscriptomes more squarely on the hydrogenase genes and their expression rather than predominantly on MAGs (see Major Comments below). The analysis of MAGs could be extended to better understand how the organisms are physiologically responding to moisture limitation. There are also interpretations related to SOC that are unclear. The authors could revise based on the feedback to make the paper suitable for publication.

- Is the methodology sound? Does the work meet the expected standards in your field?

Yes.

- Is there enough detail provided in the methods for the work to be reproduced?

No, but the missing points are minor and could be added in a revision.

Summary

Reji et al., demonstrate that H₂ oxidation occurs at some of the lowest water potentials known for microbial metabolism and revise H₂ sink models based on these results. The authors integrate these findings into a global model and show that arid regions contribute disproportionately to the global H₂ sink, which has significant implications for climate-relevant H₂ cycling. A key strength of the paper is the scaling of measurements from the microbe to global scales and integration of lab-based and modeling approaches. The determination of a new, lower water potential threshold is also a notable contribution. A weakness of the paper as presented is its interpretation of metagenomic and metatranscriptomic data is underdeveloped, with an overreliance on MAG-level analyses and limited attention to gene-specific information (hydrogenase and otherwise). Methodological clarity is also somewhat lacking, particularly in how omics replicates were handled (particularly pooling of replicates), and the global modeling results would benefit from clearer explanation findings that appear somewhat counterintuitive in the discussion. Despite these issues, the core dataset and modeling framework are strong, and the paper makes a substantial contribution to the field. We recommend major revisions with a special focus to strengthen the omics interpretation and clarify key methodological and conceptual points.

Major comments

1) A more thorough exploration of the omics data

It sounds like DNA/RNA was extracted from individual forest replicates but from pooled meadow and pine barrens soil. It needs to be explicitly stated in the methods how DNA/RNA were pooled before sequencing and whether any replicates (e.g., for the forest) were retained. It doesn't appear that they were, which means that all downstream analyses cannot include replicate-based statistical tests. Please discuss how the lack of statistical evaluation limits the interpretability of the results. There are error bars on Figure S6, but these appear to be internally derived from the stdev on the mean of n=2 genes within a given metagenome.

A primary concern is that the majority of the analysis of the genomic data are not directly related to the activity of the hydrogenase genes but are more focused on overall measures for the MAGs. Could the analysis be improved by implementing a moisture-based analysis of any of the following suggestions?

(i) Could the expression levels of Hase in the entire sample, including unbinned contigs, be plotted to show overall community responses (like Fig. S6 but for expression)?

(ii) Instead of showing the expression of the MAGs as a whole (which needs to be defined more clearly in the methods exactly what that means and how it is determined), could Fig. 3b show the patterns in expression of the Hase in those MAGs?

(iii) The paper would benefit from an exploration of comparisons between transcript levels of the H₂ oxidizing genes and the other pathways of interest. Could the relative expression of Hase vs other genes including the osmoprotectant genes (to support statements in L591) and C-cycling genes (L509-515) in Fig. 3b be evaluated on a per-MAG basis?

(iv) Overall, the patterns in Hase MAG abundance and expression were not sufficiently discussed. Given that such a high proportion of organisms appear to encode for these Hases (Fig. S6), it is unclear how much can be derived from the pathway represented in 12 MAGs (Fig. 3b), especially without comparison to the pathways represented in the other 24 MAGs recovered without Hases. To support Fig. 3a, we think it's necessary to not only look at those bins which contain the

potential for H₂ oxidation, but also those other MAGs that don't. It would support claims about the functional roles of H₂ metabolizers through comparison with the general microbial community or perhaps phylogenetically similar organisms that do not encode the H₂ metabolism trait.

(v) The relationship of these findings could be plotted versus water potential to more clearly evaluate their role in driving H₂ oxidation, as you nicely show in Figure 1b.

2) Relating H₂ oxidation to soil carbon

The discussion of the ecological strategies for H₂ oxidizers in low versus high C soils would benefit from additional clarification and organization—especially to help explain apparent contradictions. The first part of the discussion in L580 on carbon-rich soils appears to conflict with the later part in L584 on oligotrophic soil. If the H₂ uptake per unit SOC is higher in the non-forest soils, how do we come to the conclusion that there is a higher reliance of the microbial community on alternative energy sources such as H₂ in the forest soil? It could help to define whether 'higher reliance' means as a community average or a per cell average? Could the authors add data from other papers to Fig. S4b to support the linear model. With three data points distributed, it is easy to get a high R² and assume a linear form.

3) Clarification of global projections for deposition velocity

The results and discussion on the change of deposition velocity globally require additional clarification (Fig. 2). Specifically, the paper does not explain why lowering the water potential threshold (e.g., from -3 to -100 Mpa) would decrease deposition velocity in continental regions. It is unclear whether this somehow shifts the model relationships (given by equations in this MS and Fig. 2 in Bertagni et al., 2021) or through some other interaction. Physically, what could cause microbes to be able to access H₂ at even lower moisture levels but reduce the deposition velocity of H₂ overall? We understand that Fig. 2c is compositional (%; this figure would be clearer if plotting the absolute values instead of percentage data) so the decrease in continental could just be at the expense of the desert increasing. But continental values for v_{dep} are also decreasing in an absolute sense in Fig in 2b vs 2a, and this is not sufficiently explained. Does this new formulation not only allow more uptake at low moisture levels but also temper the amount of uptake at higher moisture levels too? Along these lines, the implications for the projected change in v_{dep} over time should be discussed more thoroughly.

Minor comments

We understand there are trade-offs in selecting the timeline of the experiment. Please discuss the impact of the duration of the incubations relative to the possible growth rate of these organisms in the manuscript.

Temperate forests and grasslands are some of the best represented by current data, and the authors should explain why these sites were chosen. How could the fact that your sites are all acidic influence your results and conclusions?

L44: Define pp in the abstract.

Lines 75-76: "The global warming potential of H₂" should likely also indicate that this is "indirect".

Lines 87-91: Might be beneficial to expand a bit on the work that has been done rather than essentially just saying it has led to inaccurate estimations.

L117: Ending the introduction with a statement on what the big-picture importance of the work could be would be more compelling than ending by listing another method.

L126: Since you have two forest sites, could you use more specific shorthand names rather than forest vs PB?

Lines 129-130: "At each location, soil samples were collected from the top ~10 cm, after removing the organic layer, if present." Would we expect the organic layer to not impact H₂ diffusion in natural systems? Both by abiotic (diffusion limitation) and biotic (H₂ oxidation) mechanisms.

L200: Please provide full details for your H₂ oxidation rate calculation. Please give a range of estimated fluxes for comparison to other studies. Specify what you used as the initial H₂ concentration to do so and why (one concentration for all samples or did it vary between samples as initial H₂ measurement)?

L207-209: Give soil mass to be consistent with what you stated you added to the jars. How much soil was used for the gravimetric moisture analysis? Was it enough to determine significant shifts in moisture? Describe any limitations in the water potential measurement for such a small amount of soil.

L228: Is this the right table? It is not clear from this combination of tables what the moisture levels were for the samples sequenced, unless it is all the samples in Table S1. Please clarify in the caption.

L340: Fastest microbial H₂ uptake

L363: It would be interesting to discuss these observations of the impact of recently added water to H₂ oxidation in light of the intermittent H₂ uptake concepts presented in Bertagni et al., 2021.

L463: We have seen H₂ase used more frequently (10.1111/j.1574-6976.2001.tb00587.x.) than Hase

L484: Check grammar

Fig. 1: 'Faded' colors are pretty hard to distinguish. Could you visualize in a more specific way? It's not clear initially that the AZ soils are represented by a range of bars. We think the problem is that they are located up near all the other legends. Could you differentiate the legends from the ranges, perhaps with a box around the legends? We don't think they need to have separate colors since there are no points to look for in the plot—they just add confusion. Clarify whether or not these bars relate to the y axis as well. What does '/porosity' mean in the caption? This should be placed in methods instead: Saturation levels (%) were estimated as volumetric moisture content/porosity.

Fig. 2: Is there a way that you could annotate Fig. 2d to indicate what a positive vs negative change means? We are assuming positive is an increase in v_d in the later period vs the earlier period, but we wonder if you could annotate that to be more clear? At first, we were confused whether c and d corresponded with a and b, and others might be too. It might also be useful to adjust the axis labels on the bar charts so that they are centered beneath the bars rather than far to the side of them.

Fig. 3b: Check spelling "Flgellum"

Table S1: Indicate if water content is gravimetric % saturation.

Fig. S4: H₂ oxidation rates increase with

(Remarks on code availability)

We were unable to access the code. The github link did not connect to us to a functioning page.

Reviewer #3

(Remarks to the Author)

(Remarks on code availability)

Response to Reviewer Comments

Reviewer #1 (Remarks to the Author):

Dear Authors and Editors,

Reji, Bertagni, Paulot, Qin and Zhang present new measurements combined with analysis that offers a wealth of new insight into soil microbial uptake of hydrogen. The authors correctly identify that this soil sink remains poorly constrained. They conduct new measurements that inform how the soil sink is related to the availability of soil water to microbes. The analysis then identifies contenders for hydrogen oxidising microbes present in the tested soils and, by inserting their water potential constraint into a theoretical soil uptake model, probes the planetary distribution of soil uptake by climate zones.

I consider that this work is a useful contribution to our understanding of the hydrogen soil uptake in low-moisture soils. The particular significance of their results comes from their finding that hydrogen oxidising microbes uptake hydrogen even when water is much harder to obtain from soils than the wilting point of plants. I expect that there is wide interest in how this underscores the fascinating nature of hydrogen as an important energy source for microbes in the most challenging environments. For those of us focused on the hydrogen budget, this work has important implications for how soil uptake may be included in atmospheric chemistry simulations – the authors find a more important role for uptake in arid regions – and may carry implications for policy-important hydrogen GWP100 calculations.

One disagreement between this study and an ongoing hydrogen uptake study (Drewer, Cowan, Nemitz et al., shared publicly: <https://h2envimpacts.org.uk/environmental-impacts-of-hydrogen-energy-meeting/> : <https://drive.google.com/drive/folders/1Dvm5pzbnL8D-mA3EMzFTqnEVVAKKYUKB>) is in their choice of just several grammes of soil (ln 180-210) compared with '~800 g', 'packed to roughly field bulk density', in the Drewer et al. method. Please can the authors explain why they choose such a small mass. Additionally, please can they comment on the implications for diffusive or soil-moisture dynamics of having such a relatively large surface-area to volume in these experiments compared to others.

Response: *We thank the reviewer for these insightful comments. The use of a small soil mass in our incubations was specifically employed to eliminate/minimize diffusive limitation for H₂ uptake. This is because our goal was to determine specific quantities related to HOB activity, namely the moisture threshold and the uptake rate, which carry critical implications in the biogeophysical understanding of the H₂ soil sink and bacterial community but currently lack quantitative constraints. Reducing the amount of soil was*

therefore essential to ensure that measured rates reflected potential biological activity rather than diffusion limitations. As shown in Figure S1, the concentration of H₂ in soil pores and bottle headspaces remained consistent across moisture levels in the dry range, supporting the conclusion that diffusive limitation was minimal in our incubations. This is now more clearly emphasized in the manuscript at lines 96-98, 105-106, 172-173 (revised manuscript).

Can the authors include an explicit comparison between their results and with the recent work of Paulot et al. (ACP,2024), where similar schemes with $\Psi_{ws} = -3$ MPa and -10 MPa are compared using GLDAS data instead of ERA5 hourly (ln 300)?

Response: Thank you for these comments. We used ERA5 instead of GLDAS for two reasons: (1) GLDAS generally includes higher moisture estimates, which are not necessarily realistic, especially in very arid regions. With a lower moisture threshold for bacterial uptake, as highlighted by our experiments, the GLDAS dataset would result in nearly all arid regions, including the Sahara, being active for H₂ uptake (Fig. 1a below). This raises some questions about the moisture datasets themselves, as well as on the limits of H₂ uptake in real desert environments since the Sahara Desert hosts very little microbial biomass. Constraining these will require direct field measurements. Generally, use of GLDAS results in a larger increase in $vd(H_2)$ with a lower moisture threshold of -100 MPa, while there is reasonable agreement between the two for -3 MPa as the threshold (Fig. 1b below). (2) GLDAS data begins in the 1980s, whereas ERA5 allows for a longer temporal comparison.

Figure 1: Comparing the effect of dataset choice for estimating $vd(H_2)$. (a, top panel): Fraction of time when soil moisture is above the moisture threshold (-3 MPa vs. -100 MPa) using GLDAS vs. ERA5. Most regions are above the threshold (at -100 MPa) when using GLDAS. (b, bottom panel): H_2 deposition velocity across time estimated using ERA5 vs GLDAS for two water-stress thresholds (blue: -100MPa and red: -3 MPa).

An additional analysis that would boost the importance of this work is a quantification of the sub-decadal variability in soil uptake with water-stress at -100 MPa compared with water-stress at -3 MPa. This is a small extension from their calculated 1950-2019 uptake change (ln 438-441) that would provide useful insights into a poorly understood variability of atmospheric hydrogen.

Response: Thank you for this suggestion. We have now added an analysis of the sub-decadal variability with the revised water-stress threshold (new Fig. S5). Notably, a lower moisture threshold has a stronger effect on $vd(H_2)$ on longer/interdecadal timescales rather than short-term/interannual timescales (L454-456).

Signed cordially,

Alex Tardito Chaudhri

Minor comments:

(ln 39) suggestion: these water potentials could be contextualised, e.g. ln 354,535 -3 MPa is mentioned as the wilting point for semi-arid plants.

Response: Revised the sentence as: "We report H_2 oxidizer activity down to -70 to -100 MPa water potentials across soils, which are among the driest conditions reported for microbial activity and are much drier than assumed in global simulations of H_2 ." (L 35-38).

(ln 69-70) different estimates vary from 60s%-80s% so I'm not convinced that we can

confidently state that soil uptake is certainly over 70% of sink. E.g., based on atmospheric hydrogen's isotopic signature Rhee et al. 2006 get over 80%, but based on some assumptions and what works in their models the Sand et al. 2023 estimate c.70% but with some models less than that.

Response: *Agreed, we revised to between 60-80% (L 65).*

(In 89-90) this mention of the 'paucity of observational constraints' is very important, and I am impressed that the authors seek to address this. Suggestion: however, there is a relative abundance of measurements in temperate NH America, so is it worth highlighting the particular paucity of soil flux measurements in (semi-)arid soils?

Response: *Revised to "particularly in moisture-limited, arid, and semi-arid soils" (L 86).*

(In 126,127,174) there is a lack of consistency in lat-lon reporting. For usability, I would suggest reporting in degrees with decimals in both cases, and for decimal places is it important or accurate that a location is reported to a precision equivalent to one metre?

Response: *We thank the reviewer for highlighting this inconsistency in lat-lon reporting. We have revised the degrees in decimals, as suggested. The precision to five decimal places (~1 meter) is not critical, but it approximately reflects the spatial scale of subsampling, as at each site, soil samples were collected from three sub-locations spaced approximately 1 meter apart (see Methods L 125, 127, and 128).*

(In 126,127) reliability: are there issues with collecting samples in single locations in your sample areas? I understand that in-situ flux chamber measurements show enormous spatial variability uptake within metres of each other (e.g. Cowen et al. EGU sphere preprint-2024).

Response: *Spatial variability is indeed a major axis of variability, and we thank the reviewer for drawing attention to a methodological detail that was omitted from the previous version. We have now clarified our sampling strategy to specify that at each location, three locations spaced approximately 1 m apart were sampled and combined into a composite sample (L 128-133).*

(In 154) hyphen formatted as minus sign.

Response: *Fixed to "combusted weight" (L 156).*

(In 196-198) suggest these rates would be clearer to understand in ppb/min/g (to check – sometimes units formatted gSoil, other times as g-Soil).

Response: *Revised as suggested.*

(In 198-200) is this as trivial as discussed here? Chamber uptake of hydrogen is seen to occur on very short timescales, please explain how this model of drawdown is robust.

Response: *The signal in the decaying H₂ concentration acts on a timescale that depends on the relative ratio between available substrate (scales with head space*

volume) and bacterial activity (soil volume and uptake rate). We specifically designed our experiments to employ a limited amount of soil volume that would allow us to easily discriminate against an exponential decay of the H₂ concentration within tens of minutes. We stress that an exponential decay, in addition of being confirmed by experimental results, has to be expected, as the sink is proportional to the H₂ concentration present ($VdC/dt=k C$, leading to an exponential solution for C). Although sometimes a linear uptake is referred to in the literature, there is no evidence of such a decay trend in the concentration, which would require an unrealistic constant sink, independent of the concentration. Chamber uptake is indeed faster, especially since the chamber volume includes a much larger quantity of soil (relative to chamber headspace volume) than what we had in our incubations. Furthermore, our incubations also isolated the biotic component of uptake by minimizing physical diffusive limitation as would be present in a chamber experiment in the field, so data from these two experimental systems are not directly comparable. A first-order approximation was deemed appropriate since the uptake curves generally showed an exponential decay trend (Fig. S3).

(In 215-266) I indicate that these sections are beyond my expertise. Suggestion: is it worth commenting how widely established these methods to give context to readers without this expertise?

Response: We have added a sentence to clarify these methods are standard and widely established (L 213-215).

(In 317-318) suggestion: considering your helpful phrasing about the intrinsic nature of Ψ (In 354-355,379-381), could the link be better explained here? Also, do your ideas in In 550-553 not give some good justification for why you might want to make this link sooner?

Response: Thank you, we revised the original L317-318 as: “we then show that using the soil water potential rather than soil moisture offers better quantitative insights on the shared water-stress limitations for HA-HOB across soil types, as water potential provides a more intrinsic measure of water availability that is comparable across different soil textures” (L 329-333).

(Fig. 1) are the scattered points the same tests in each panel, can the triangles, circles, squares in a also be used in b. The legend in b shouldn't have the scatter marker lying on a line.

Response: We apologize for any confusion. In response to this comment, as well as a similar comment from reviewers 2 and 3, we revised Fig. 1 to remove the shapes and use only colors to differentiate between soil types.

(In 434-436) please see commentary above regarding quantifying sub-decadal variability of uptake.

Response: *We have added a new figure to compare sub-decadal variability (Fig. S5).*

(In 477) typo 'to be range from'

Response: *corrected (L 500).*

(In 484) indicating

Response: *corrected (L 547).*

(In 498-499) suggestion: is it worth commenting on how usual/unusual it is to find a novel genus in such tests, e.g. is it convincing that it is a novel genus rather than a measurement error?

Response: *It isn't unusual to detect novel genera or even higher-level taxa in soil samples, as a significant fraction of microbial diversity remains uncharacterized (often termed the "microbial dark matter"). Taxonomic classification is not subject to typical measurement errors encountered in analytical methods but is based on DNA sequence identity and comparison against established taxonomic similarity thresholds. Our classification is based on well-accepted bioinformatic programs and reference databases (GTDB-tk), providing robust support for identifying a novel genus rather than a technical artifact.*

(Fig. 3) Clarity: as each genus can only be in one phylum please can the legend be simplified to list the genera under sub-headings for the phyla with the correct symbols (currently they all look like acidobacteriota in the legend and it is hard to determine colours). Then can you omit the 'p__'s and 'g__'s? Is 'Flagellum' a misspelling and should 'flagellin' have a capital F?

Response: *We appreciate these detailed suggestions. While the prefixes are standard practice in microbial ecology to clarify taxonomic rank and avoid ambiguity, we have revised the figure to remove them in order to improve accessibility for a broader audience. We have also copy-edited the figure as suggested (Fig. 4).*

(In 576-578) this might be true in the sub-tropics and tropics, and presumably also when you integrate everywhere, but can be potentially misleading if soil moisture is not the primary driver everywhere. In (23) and (26), $h(T)$ has a steep gradient between 0 and 20 degC. (35) see that the observed hydrogen distribution can be explained where NH extra-tropics mostly respond to temperature seasonality.

Response: *We appreciate this comment and interpret this as a suggestion to highlight that moisture may not be the primary control on H_2 uptake across biomes. We revised the sentence to "soil moisture influences both the microbial activity and the physical diffusion of H_2 through the soil matrix to remain the major modulator of H_2 soil uptake."*

Reviewer #1 (Remarks on code availability):

https://github.com/Linta-Reji/Reji2025_H2_SoilMoisture provides clear instructions and data files are available. (I have searched through the repository but not have not run the code in R).

Response: *Thank you!*

Reviewer #2 (Remarks to the Author):

Journal-specific considerations

- What are the noteworthy results?

This article is the first to systematically relate soil H₂ uptake to water potential, and the authors show that H₂ oxidation by microbes occurs at some of the lowest water potentials of any known microbial process. The authors implement new understanding of the relationship of H₂ uptake and soil water-potential into a global model, which shows that arid lands likely make a relatively larger contribution to the global H₂ sink

- Will the work be of significance to the field and related fields? How does it compare to the established literature? If the work is not original, please provide relevant references. Yes, this work represents a significant advancement to our understanding of the moisture sensitivity of overwhelming soil sink for atmospheric H₂. The field is limited in data, and this systematic study that was well-designed to meet the needs for upscaling with models makes an important contribution to our understanding of its major driver of variability.

- Does the work support the conclusions and claims, or is additional evidence needed? Yes.

- Are there any flaws in the data analysis, interpretation and conclusions? - Do these prohibit publication or require revision?

We suggest that the authors focus their analysis of metagenomes and metatranscriptomes more squarely on the hydrogenase genes and their expression rather than predominantly on MAGs (see Major Comments below). The analysis of MAGs could be extended to better understand how the organisms are physiologically responding to moisture limitation. There are also interpretations related to SOC that are unclear. The authors could revise based on the feedback to make the paper suitable for publication.

Response: *We thank the reviewer for their work and appreciate these suggestions and have revised the manuscript as outlined in response to specific comments below.*

- Is the methodology sound? Does the work meet the expected standards in your field?

Yes.

- Is there enough detail provided in the methods for the work to be reproduced?
No, but the missing points are minor and could be added in a revision.

Summary

Reji et al., demonstrate that H₂ oxidation occurs at some of the lowest water potentials known for microbial metabolism and revise H₂ sink models based on these results. The authors integrate these findings into a global model and show that arid regions contribute disproportionately to the global H₂ sink, which has significant implications for climate-relevant H₂ cycling. A key strength of the paper is the scaling of measurements from the microbe to global scales and integration of lab-based and modeling approaches. The determination of a new, lower water potential threshold is also a notable contribution. A weakness of the paper as presented is its interpretation of metagenomic and metatranscriptomic data is underdeveloped, with an overreliance on MAG-level analyses and limited attention to gene-specific information (hydrogenase and otherwise). Methodological clarity is also somewhat lacking, particularly in how omics replicates were handled (particularly pooling of replicates), and the global modeling results would benefit from clearer explanation findings that appear somewhat counterintuitive in the discussion. Despite these issues, the core dataset and modeling framework are strong, and the paper makes a substantial contribution to the field. We recommend major revisions with a special focus to strengthen the omics interpretation and clarify key methodological and conceptual points.

Major comments

1) A more thorough exploration of the omics data

It sounds like DNA/RNA was extracted from individual forest replicates but from pooled meadow and pine barrens soil. It needs to be explicitly stated in the methods how DNA/RNA were pooled before sequencing and whether any replicates (e.g., for the forest) were retained. It doesn't appear that they were, which means that all downstream analyses cannot include replicate-based statistical tests. Please discuss how the lack of statistical evaluation limits the interpretability of the results. There are error bars on Figure S6, but these appear to be internally derived from the stdev on the mean of n=2 genes within a given metagenome.

Response: *Thank you for these comments. As noted in the original Methods (Lines 218–222 of the previous version), DNA and RNA from the meadow soil and Pine Barrens sand were pooled prior to sequencing to ensure sufficient nucleic acid yield for metagenome and metatranscriptome sequencing. Forest soil samples, by contrast, yielded sufficient nucleic acids from individual replicates, so one replicate each per moisture level were sequenced separately. We have now re-stated this information in Results (L 485–487), in addition to in Methods (L219–223), to improve transparency. We are also including a new table (Table S3) that summarizes the extraction strategy.*

We acknowledge that pooling of samples prior to extraction/sequencing precludes replicate-based statistics, and accordingly, statistical tests were not applied to these datasets. We believe the general trends we see across moisture levels are still robust (similar relative abundances of high-affinity H₂ases across moisture levels in each soil as shown in Fig. S7; and moisture-dependent expression of H₂ase subclusters as shown in the new Fig. 3). The relative abundances are also comparable to previous studies in forest soils (Lines 498-501). The reviewers are correct that the error bars in Fig. S6 (now Fig. S7) are not based on replicate-based statistics. While this information was included in the original caption, we apologize if it was unclear. We have now added a new sentence to the figure caption to explicitly state that the error bars do not represent biological replicates (L 117-118, SI).

A primary concern is that the majority of the analysis of the genomic data are not directly related to the activity of the hydrogenase genes but are more focused on overall measures for the MAGs. Could the analysis be improved by implementing a moisture-based analysis of any of the following suggestions?

(i) Could the expression levels of Hase in the entire sample, including unbinned contigs, be plotted to show overall community responses (like Fig. S6 but for expression)?

Response: *This is a good suggestion—we have now added a new figure (Fig. 3) on overall HA-H₂ase expression levels, in addition to MAG-based analyses.*

(ii) Instead of showing the expression of the MAGs as a whole (which needs to be defined more clearly in the methods exactly what that means and how it is determined), could Fig. 3b show the patterns in expression of the Hase in those MAGs?

Response: *Thank you for this suggestion. Panel b now shows relative expression of the hydrogenases in these MAGs (new Fig. 4).*

Regarding clarity on methods: Panel b originally showed the relative transcriptional activity of the MAGs—which essentially translates to the relative number of RNA reads that are mapped to each MAG. These were estimated by using standard methods as described in the original Methods (L251-257 in the original manuscript), which involved mapping metatranscriptomic reads against each MAG using the program Bowtie2, and then normalizing the counts of mapped reads as “transcripts per million” (L268-272). We agree that showing the expression of HA-H₂ases is more relevant to the story, and therefore, the figure has been revised as such (Fig. 4).

(iii) The paper would benefit from an exploration of comparisons between transcript levels of the H₂ oxidizing genes and the other pathways of interest. Could the relative expression of Hase vs other genes including the osmoprotectant genes (to support statements in L591) and C-cycling genes (L509-515) in Fig. 3b be evaluated on a per-MAG basis?

Response: *We appreciate this thoughtful suggestion; however, we do not think this comparison is essential for the interpretations presented. There is no expectation of*

coordinated expression among these genes; rather, the collective presence of these pathways supports the interpretation that these organisms occupy an oligotrophic niche. High-affinity hydrogen oxidation is among the several stress adaptation strategies that these organisms appear to possess.

(iv) Overall, the patterns in Hase MAG abundance and expression were not sufficiently discussed. Given that such a high proportion of organisms appear to encode for these Hases (Fig. S6), it is unclear how much can be derived from the pathway represented in 12 MAGs (Fig. 3b), especially without comparison to the pathways represented in the other 24 MAGs recovered without Hases. To support Fig. 3a, we think it's necessary to not only look at those bins which contain the potential for H₂ oxidation, but also those other MAGs that don't. It would support claims about the functional roles of H₂ metabolizers through comparison with the general microbial community or perhaps phylogenetically similar organisms that do not encode the H₂ metabolism trait.

Response: *Thank you for these suggestions. To panel (b) (now revised Fig. 4), we have now added an additional feature that summarizes the prevalence of a given pathway group in the non-HOB MAGs.*

(v) The relationship of these findings could be plotted versus water potential to more clearly evaluate their role in driving H₂ oxidation, as you nicely show in Figure 1b.

Response: *Thank you for this suggestion. Since the relationship between microbial omics results and water saturation/potential is qualitative, we think plotting these against water potential as opposed to saturation would offer the same results.*

2) Relating H₂ oxidation to soil carbon

The discussion of the ecological strategies for H₂ oxidizers in low versus high C soils would benefit from additional clarification and organization—especially to help explain apparent contradictions. The first part of the discussion in L580 on carbon-rich soils appears to conflict with the later part in L584 on oligotrophic soil. If the H₂ uptake per unit SOC is higher in the non-forest soils, how do we come to the conclusion that there is a higher reliance of the microbial community on alternative energy sources such as H₂ in the forest soil? It could help to define whether 'higher reliance' means as a community average or a per cell average? Could the authors add data from other papers to Fig. S4b to support the linear model. With three data points distributed, it is easy to get a high R² and assume a linear form.

Response: *We thank the reviewer for drawing attention to this. To clarify, when we refer to an 'oligotrophic niche,' we mean the metabolic lifestyle of these microbes—adapted to utilize low-resource substrates—rather than the carbon content of the soil itself (i.e., "oligotrophic niche" does not equate to "oligotrophic soil"). These HA-HOB are mixotrophic, capable of oxidizing H₂ even in carbon-rich soils where competition for*

organic carbon may be intense. We have revised the text to make this distinction clearer (L 659-661).

*The reviewers are also right to point out that a linear model is poorly supported with only three data points. We acknowledge this limitation, which is why simulations with SOC modulation are provided as supplementary figures. Such linear relationships have been observed in prior studies (e.g., Hou et al., 2025, <https://doi.org/10.1016/j.soilbio.2025.109831>; Popelier et al., 1985, *Plant and Soil* 85, 85 – 96). A key limitation in incorporating data from other papers to our model though is the fact that the rates obtained in previous work are not directly comparable to our measurements (diffusive limitation likely present in previous work). We also understand more complex parameterizations involving soil carbon are being developed, and emphasize the need to better unravel the mechanistic links between SOC and H₂ oxidation rates (L 644-646).*

3) Clarification of global projections for deposition velocity

The results and discussion on the change of deposition velocity globally require additional clarification (Fig. 2). Specifically, the paper does not explain why lowering the water potential threshold (e.g., from -3 to -100 Mpa) would decrease deposition velocity in continental regions. It is unclear whether this somehow shifts the model relationships (given by equations in this MS and Fig. 2 in Bertagni et al., 2021) or through some other interaction. Physically, what could cause microbes to be able to access H₂ at even lower moisture levels but reduce the deposition velocity of H₂ overall? We understand that Fig. 2c is compositional (%; this figure would be clearer if plotting the absolute values instead of percentage data) so the decrease in continental could just be at the expense of the desert increasing. But continental values for v_{dep} are also decreasing in an absolute sense in Fig in 2b vs 2a, and this is not sufficiently explained. Does this new formulation not only allow more uptake at low moisture levels but also temper the amount of uptake at higher moisture levels too? Along these lines, the implications for the projected change in v_{dep} over time should be discussed more thoroughly.

Response: *Thank you for drawing attention to this. As detailed in the Methods section, $v_d(H_2)$ is optimized for each configuration, such that the global deposition average $v_d(H_2)$ is 0.09 mm/s for year 2010. Thus, our focus is primarily on the relative contribution of different climate zones to the H₂ sink. In particular, a lower water potential threshold for HA-HOB makes more land available in dry regions, which has to be compensated by a reduction in $v_d(H_2)$ in regions that are generally not limited by dry conditions (e.g., continental regions).*

Minor comments

We understand there are trade-offs in selecting the timeline of the experiment. Please discuss the impact of the duration of the incubations relative to the possible growth rate of these organisms in the manuscript.

Response: We agree that the duration of incubation is an important consideration when measuring microbial process rates. Our incubations lasted 5–6 days, which we selected to ensure stable measurement of H_2 oxidation potential while minimizing large community shifts. Additionally, soil bacterial growth rates are generally low, with in situ doubling times often on the order of days to weeks—a recent analysis showed average generation times of 14 to 45 d, with a range of 4 to 402 days (Caro et al., 2023). Even these estimates reflect optimal laboratory conditions rather than field-relevant rates. Thus, while we cannot rule out the possibility of some microbial growth during the incubation, we expect that the measured H_2 oxidation rates primarily reflect the metabolic potential of the original community rather than substantial enrichment of H_2 oxidizers during incubation.

Caro, T. A., et al., 2023, PNAS, 120 (16) e2211625120.
<https://doi.org/10.1073/pnas.2211625120>

Temperate forests and grasslands are some of the best represented by current data, and the authors should explain why these sites were chosen. How could the fact that your sites are all acidic influence your results and conclusions?

Response: We agree with the reviewer that temperate forests and grasslands are relatively better studied biomes in prior H_2 oxidation research. Our site selection, however, was guided by the study's central question—identifying the soil moisture threshold for high-affinity hydrogen oxidizers—rather than by an aim to maximize coverage of global soil types. All of our sites are acidic, which may influence community composition and activity. We have added a statement in the Discussion acknowledging the need to characterize more soil types (L 688-690).

L44: Define pp in the abstract.

Response: Revised.

Lines 75-76: “The global warming potential of H_2 ” should likely also indicate that this is “indirect”.

Response: Revised as suggested (L 72).

Lines 87-91: Might be beneficial to expand a bit on the work that has been done rather than essentially just saying it has lead to inaccurate estimations.

Response: Thank you for this suggestion. We have revised this sentence to clarify that limitations in prior work primarily stem from sparse data in moisture-limited, arid, and semi-arid soils. The revised sentence now reads: “Due to the paucity of observational constraints on soil uptake, particularly in moisture-limited, arid, and semi-arid soils, these parameterizations result in inaccurate estimates of the seasonality and spatial variability of uptake” (L 85-88).

L117: Ending the introduction with a statement on what the big-picture importance of the work could be would be more compelling than ending by listing another method.

Response: resolved (L 113-115).

L126: Since you have two forest sites, could you use more specific shorthand names rather than forest vs PB?

Response: *We thank the reviewers for this suggestion. While we have two forested sites, these are not equivalent at all—the first is a typical deciduous temperate forest while the Pine Barrens represents a highly nutrient-poor sandy ecosystem distinct from typical forests. We therefore prefer to retain the terms ‘forest’ and ‘Pine Barrens’ throughout the text and figures to highlight this ecological contrast, while providing site details in the Methods for clarity.*

Lines 129-130: “At each location, soil samples were collected from the top ~10 cm, after removing the organic layer, if present.” Would we expect the organic layer to not impact H₂ diffusion in natural systems? Both by abiotic (diffusion limitation) and biotic (H₂ oxidation) mechanisms.

Response: *Indeed, the organic (litter) layer potentially has a significant influence on H₂ oxidation. However, our goal here is to not assess natural variability in H₂ oxidation rates, rather to isolate the effect of soil moisture on the biotic process within the mineral soil. Including the organic layer would have introduced additional variability due to differences in litter composition, depth, and moisture-holding capacity among sites, making it more difficult to disentangle the specific role of soil moisture on HA-HOB activity.*

L200: Please provide full details for your H₂ oxidation rate calculation. Please give a range of estimated fluxes for comparison to other studies. Specify what you used as the initial H₂ concentration to do so and why (one concentration for all samples or did it vary between samples as initial H₂ measurement)?

Response: *All the measured concentrations and the code we used to determine oxidation rates are included in the specified GitHub repository. As specified in the Methods, H₂ oxidation was modeled as a first-order decay process, and the rate constants (min⁻¹) were used as a proxy for oxidation rates. The initial H₂ concentration varied across incubations as we are starting the experiment with ambient H₂ levels each time. The calculated rate constants therefore reflect the removal rate at these near-ambient concentrations.*

To emphasize this further, we have added a new sentence to Methods: “Initial H₂ concentrations matched the corresponding ambient H₂ levels measured for each experiment.” (L 195-197).

L207-209: Give soil mass to be consistent with what you stated you added to the jars. How much soil was used for the gravimetric moisture analysis? Was it enough to

determine significant shifts in moisture? Describe any limitations in the water potential measurement for such a small amount of soil.

Response: *We apologize for any confusion/lack of clarity and have now specified in the Methods the exact amounts of soil used for gravimetric moisture determination. For water potential measurements, the WP4C sample cups are filled halfway for analysis as the protocol requires, which corresponds to ~1-2 mL of soil (volume and weight varies with soil type/texture). This is the standard sample amount recommended by the manufacturer for WP4C measurements. Although fine-scale heterogeneity in soil structure/moisture could indeed affect both gravimetric and WP measurements, our values were generally consistent among replicates within each soil-moisture level combination.*

L228: Is this the right table? It is not clear from this combination of tables what the moisture levels were for the samples sequenced, unless it is all the samples in Table S1. Please clarify in the caption.

Response: *Added a new SI table summarizing extraction results. (Table S3)*

L340: Fastest microbial H₂ uptake

Response: *Revised as suggested (L355-356).*

L363: It would be interesting to discuss these observations of the impact of recently added water to H₂ oxidation in light of the intermittent H₂ uptake concepts presented in Bertagni et al., 2021.

Response: *Thank you for this insightful comment. Our results indeed suggest that the water-stress threshold for soil bacteria may be lower than previously assumed. This implies that intermittent H₂ uptake could still occur, but with a lower moisture threshold—potentially one that can be reached even with morning dew events. This is a particularly interesting aspect that would warrant dedicated field studies or controlled dry-wetting experiments, which, however, fall beyond the scope of the present work.*

L463: We have seen H₂ase used more frequently (10.1111/j.1574-6976.2001.tb00587.x.) than Hase

Response: *revised as suggested.*

L484: Check grammar

Response: *Corrected*

Fig. 1: 'Faded' colors are pretty hard to distinguish. Could you visualize in a more specific way? It's not clear initially that the AZ soils are represented by a range of bars. We think the problem is that they are located up near all the other legends. Could you differentiate the legends from the ranges, perhaps with a box around the legends? We don't think they need to have separate colors since there are no points to look for in the

plot—they just add confusion. Clarify whether or not these bars relate to the y axis as well. What does 'porosity' mean in the caption? This should be placed in methods instead: Saturation levels (%) were estimated as volumetric moisture content/porosity.

Response:

We made several changes to Fig. 1 to improve visual clarity:

- 1. The shaded points are now a much lighter shade.*
- 2. We added boxes around the legends*
- 3. We moved the AZ soil water potential ranges to below the horizontal axis to (a) differentiate them better from the AZ soil ranges and (b) clarify that they are not related to the vertical axis.*
- 4. In Fig. 1a, we retained the point colors, but removed the varying shapes, since they were redundant.*
- 5. We removed the sentence about saturation and porosity, as suggested (this is already discussed in Methods).*

Fig. 2: Is there a way that you could annotate Fig. 2d to indicate what a positive vs negative change means? We are assuming positive is an increase in v_d in the later period vs the earlier period, but we wonder if you could annotate that to be more clear? At first, we were confused whether c and d corresponded with a and b, and others might be too. It might also be useful to adjust the axis labels on the bar charts so that they are centered beneath the bars rather than far to the side of them.

Response: *We have revised Fig. 2 to address these comments.*

Fig. 3b: Check spelling "Flgellum"

Response: *Revised.*

Table S1: Indicate if water content is gravimetric % saturation.

Response: *Clarified as suggested.*

Fig. S4: H₂ oxidation rates increase with

Response: *Corrected.*

Reviewer #2 (Remarks on code availability):

We were unable to access the code. The github link did not connect to us to a functioning page.

Response: *Thank you for pointing this out. The first reviewer was able to access the code, so we are not sure what went wrong in this case. The code and associated data files should be available at the link provided in the Data Availability Statement.*

Reviewer #3 (Remarks to the Author):

I co-reviewed this manuscript with one of the reviewers who provided the listed reports.

This is part of the Nature Communications initiative to facilitate training in peer review and to provide appropriate recognition for Early Career Researchers who co-review manuscripts.

Response: *Thank you!*